# *Bacillus velezensis* HBXN2020 alleviates *Salmonella* Typhimurium infection in mice by improving intestinal barrier integrity and reducing inflammation

Linkang Wang[1,2,3], Haiyan Wang[1,2,3], Xinxin Li[1,2,3], Mengyuan Zhu[1,2,3], Dongyang Gao[1,2,3], Dayue Hu[1,2,3], Zhixuan Xiong[1,2,3], Xiangmin Li[1,2,3]*, Ping Qian[1,2,3]*

[1]National Key Laboratory of Agricultural Microbiology, Hubei Hongshan Laboratory, Huazhong Agricultural University, Wuhan, China; [2]College of Veterinary Medicine, Huazhong Agricultural University, Wuhan, China; [3]Key Laboratory of Preventive Veterinary Medicine in Hubei Province, the Cooperative Innovation Center for Sustainable Pig Production, Wuhan, China

**\*For correspondence:**
lixiangmin@mail.hzau.edu.cn
(XL);
qianp@mail.hzau.edu.cn (PQ)

**Competing interest:** The authors declare that no competing interests exist.

## eLife Assessment

In this **useful** study, Wang and colleagues investigate the potential probiotic effects of Bacillus velezensis in a murine model. They provide **convincing** evidence that B. velezensis limits the growth of *Salmonella* typhimurium in lab culture and in mice, together with beneficial effects on the microbiota. The overall presentation of the manuscript has improved and the work will be of interest to infectious disease researchers.

**Abstract** *Bacillus velezensis* is a species of *Bacillus* that has been widely investigated because of its broad-spectrum antimicrobial activity. However, most studies on *B. velezensis* have focused on the biocontrol of plant diseases, with few reports on antagonizing *Salmonella* Typhimurium infections. In this investigation, it was discovered that *B. velezensis* HBXN2020, which was isolated from healthy black pigs, possessed strong anti-stress and broad-spectrum antibacterial activity. Importantly, *B. velezensis* HBXN2020 did not cause any adverse side effects in mice when administered at various doses ($1\times10^7$, $1\times10^8$, and $1\times10^9$ CFU) for 14 days. Supplementing *B. velezensis* HBXN2020 spores, either as a curative or preventive measure, dramatically reduced the levels of *S.* Typhimurium ATCC14028 in the mice's feces, ileum, cecum, and colon, as well as the disease activity index (DAI), in a model of infection caused by this pathogen in mice. Additionally, supplementing *B. velezensis* HBXN2020 spores significantly regulated cytokine levels (*Tnfa*, *Il1b*, *Il6*, and *Il10*) and maintained the expression of tight junction proteins and mucin protein. Most importantly, adding *B. velezensis* HBXN2020 spores to the colonic microbiota improved its stability and increased the amount of beneficial bacteria (*Lactobacillus* and *Akkermansia*). All together, *B. velezensis* HBXN2020 can improve intestinal microbiota stability and barrier integrity and reduce inflammation to help treat infection by *S.* Typhimurium.

## Introduction

*Salmonella* Typhimurium (*S.* Typhimurium) is a major foodborne zoonotic pathogen that can cause diarrhea and colitis in humans and animals (*Fàbrega and Vila, 2013*; *Yang et al., 2021*). According to reports, *Salmonella* infections are common in both industrialized and developing nations, posing a serious risk to public health and resulting in significant financial losses (*Katiyo et al., 2019*; *Mohan et al., 2019*). At present, antibiotics remain one of the most effective treatment strategies for *Salmonella* infections. However, numerous studies have reported that the prolonged use or misuse of antibiotics can lead to environmental pollution, an increase in multi-drug-resistant bacteria, as well as gastrointestinal microbiota dysbiosis (*Ferri et al., 2017*; *Paulson and Zaoutis, 2015*; *Reyman et al., 2022*). In order to tackle *Salmonella* infections, a hunt for novel antimicrobial drugs is now underway.

More research has recently demonstrated the critical role gut microbiota plays in reducing intestinal inflammation (*Cristofori et al., 2021*) and mending the intestinal barrier (*Ohland and Macnaughton, 2010*; *Wang et al., 2020*). The use of probiotics is a popular approach to modulating intestinal microbiota nowadays. Among all probiotics, *Bacillus* is one of the most popular probiotic species because of its ability to form endospores that survive gastric transit, storage, and delivery conditions (*Gu et al., 2015*). Furthermore, *Bacillus* species can generate extracellular enzymes and antimicrobial metabolites that inhibit enteric pathogens, which lowers the risk of infection and enhances nutrient utilization (*Abdhul et al., 2015*; *Ouattara et al., 2017*).

Recent studies have found that *Bacillus velezensis* (*B. velezensis*) has the enormous potential to produce a variety of metabolites with broad-spectrum antibacterial activity (*Ye et al., 2018*). Meanwhile, previous studies have reported that *B. velezensis* exhibits varying degrees of beneficial effects on plants, livestock, poultry and fish. For instance, the combined use of *B. velezensis* SQR9 mutant and FZB42 improved root colonization and the production of secondary metabolites that are beneficial to cucumber (*Shao et al., 2022*). *B. velezensis* isolated from yaks was shown to enhance growth performance and ameliorate blood parameters related to inflammation and immunity in mice (*Li et al., 2019*). The dietary *B. velezensis* supplementation can regulate the innate immune response in the intestine of crucian carp and reduce the degree of intestinal inflammation damage induced by *Aeromonas veronii* (*Zhang et al., 2019*). However, studies on *B. velezensis* in the prevention of *S.* Typhimurium infection are rarely reported; instead, the majority of investigations on *B. velezensis* focused on the biocontrol of fungal infections in plants. In this study, a strain of *B. velezensis* HBXN2020 with broad-spectrum antibacterial activity against *Salmonella* was selected from a vast number of *Bacillus* strains. Next, we evaluated its safety in mice by supplementing *B. velezensis* HBXN2020 and then explored the protective effects in STm-infected mice. Notably, *B. velezensis* HBXN2020 alleviated colon tissue damage caused by STm infection, as indicated by markers such as STm loads, *Tnfa* and *Tjp1* levels. Moreover, supplementing *B. velezensis* HBXN2020 also increased the abundance of beneficial bacteria, specifically *Lactobacillus* and *Akkermansia*, within the colon microbiota. As a result, this research supports the creation of probiotic-based microbial products as a substitute method of preventing *Salmonella* infections.

## Results

### Isolation of *Bacillus*

Four strains of *Bacillus* with antibacterial activity were screened from a large amount of presumptive *Bacillus* isolates, based on spot-on-plate tests analysis (*Figure 1—figure supplement 1*), and the four strains clearly displayed the properties of bacteria in the genus *Bacillus* such as colonies were crateriform, Gram-positive, rod shape, and endospore-forming ability. Next, a comparative analysis of the antibacterial spectrum of four strains of *Bacillus* found that one strain exhibited excellent antibacterial activity against common pathogens (*Supplementary file 1*). Therefore, this strain was selected in this study for further research, which was designated as the HBXN2020.

### Growth curve and in vitro resistance against environmental assaults

The growth of HBXN2020 was assessed in flat-bottomed 100-well microtiter plates by measuring the values of $OD_{600}$ every hour using an automatic growth curve analyzer. After measurement, we found that HBXN2020 entered the logarithmic growth phase after 2 hr, and reached a plateau after 10 hr of culture (*Figure 1A*, *Figure 1—source data 1*).

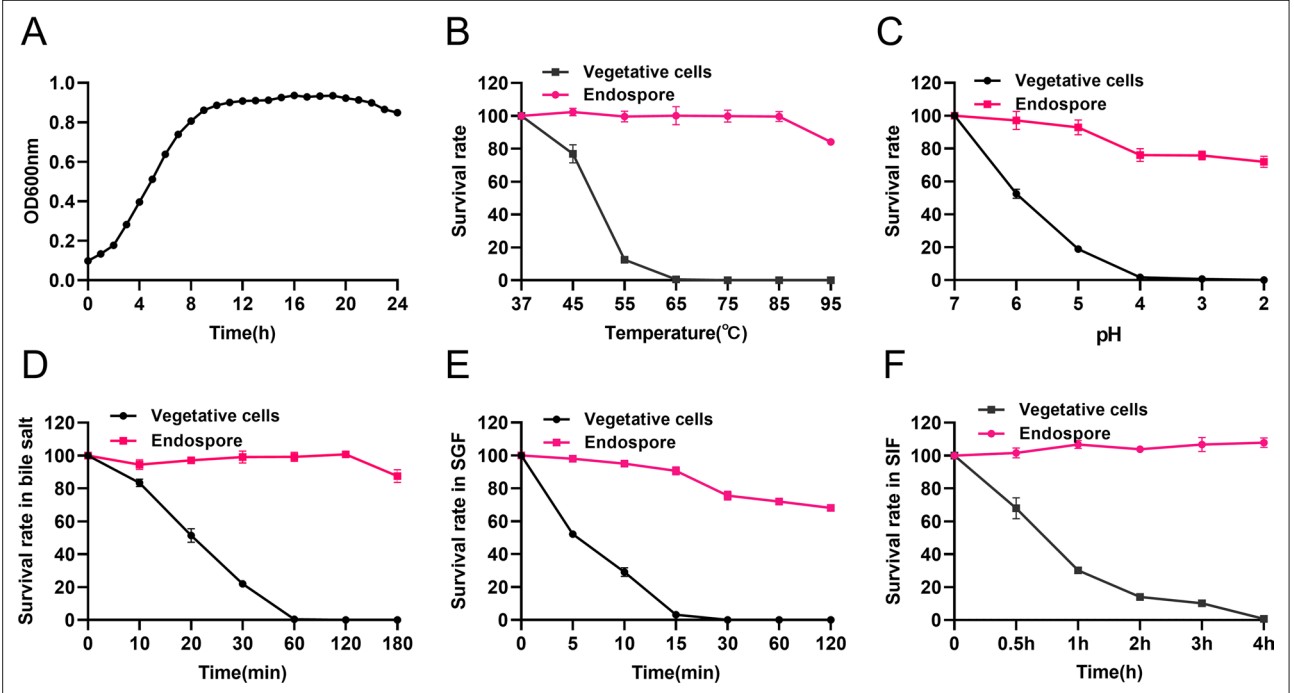

**Figure 1.** Growth curve of *B. velezensis* HBXN2020 and its in vitro resistance against environmental assaults. (**A**) Growth curves of *B. velezensis* HBXN2020 cultured in Luria-Bertani (LB) medium at 37°C, detection of $OD_{600}$ values at 1 hr intervals in microplate reader. (**B**) Survival rate of endospore and vegetative cells of *B. velezensis* HBXN2020 after 30 min at different temperatures (37°C, 45°C, 55°C, 65°C, 75°C, 85°C, or 95°C). Equal amounts of endospore and vegetative cells of HBXN2020 were exposed to the following: (**C**) acid solution (pH 2–7), (**D**) 0.3% bile salts, (**E**) simulated gastric fluid (SGF) (pH 1.2) supplemented with pepsin, and (**F**) simulated intestinal fluid (SIF) (pH 6.8) containing trypsin at 37°C. At predetermined time points, 100 µL was taken from each sample, and 10-fold serially diluted with sterile PBS (pH 7.2), and then spread on LB agar plates and cultured at 37°C for 12 hr before bacterial counting. Each group was repeated three times (n=3).

The online version of this article includes the following source data and figure supplement(s) for figure 1:

**Source data 1.** Raw data values for *Figure 1A–F*.

**Figure supplement 1.** Spot-on-plate assay of *Bacillus*.

**Figure supplement 1—source data 1.** PDF file containing original *Bacillus* antibacterial screening images for *Figure 1—figure supplement 1*.

**Figure supplement 1—source data 2.** Original files for antibacterial screening of *Bacillus* displayed in *Figure 1—figure supplement 1*.

Next, we evaluated the effect of physical, chemical, and biological sterilization conditions such as high temperature, strong acidity, and enzyme preparation on the survival of HBXN2020. The survival rate of HBXN2020 showed a decreasing trend with increasing temperature (*Figure 1B*, *Figure 1—source data 1*), but this decrease was not obvious. In a strong acid environment (pH 2.0), HBXN2020 maintained a high survival rate (*Figure 1C*, *Figure 1—source data 1*), suggesting that HBXN2020 spores can survive under extreme conditions. Based on these results, we hypothesized that HBXN2020 spores might also exhibit improved survival in gastrointestinal tract environments. Therefore, we further evaluated the survival rate of HBXN2020 in bile salt (0.3%, vol/vol), simulated gastric fluid (pepsin 1 mg/mL, pH 1.2), and simulated intestinal fluid (trypsin 1 mg/mL, pH 6.8). As shown in *Figure 1D–F* (*Figure 1D–F*, *Figure 1—source data 1*), HBXN2020 spores demonstrated significantly improved tolerance to these simulated gastrointestinal tract environments.

## Antibiotic susceptibility of HBXN2020 and bacteriostasis effect in vitro

In order to assess HBXN2020's drug resistance, 19 antibiotics that are frequently used in clinical settings were chosen in this study. The results indicated that only polymyxin B had a relatively small inhibition zone diameter (less than 15 mm). Ampicillin, meropenem, minocycline, ofloxacin, and trimethoprim had the strongest inhibition on HBXN2020, with an inhibition zone diameter exceeding 30 mm (*Figure 2A*, *Figure 2—source data 1*). This indicated that HBXN2020 was extremely sensitive to β-lactams, tetracyclines, and quinolone drugs. We next utilized an in vitro antibacterial assay to

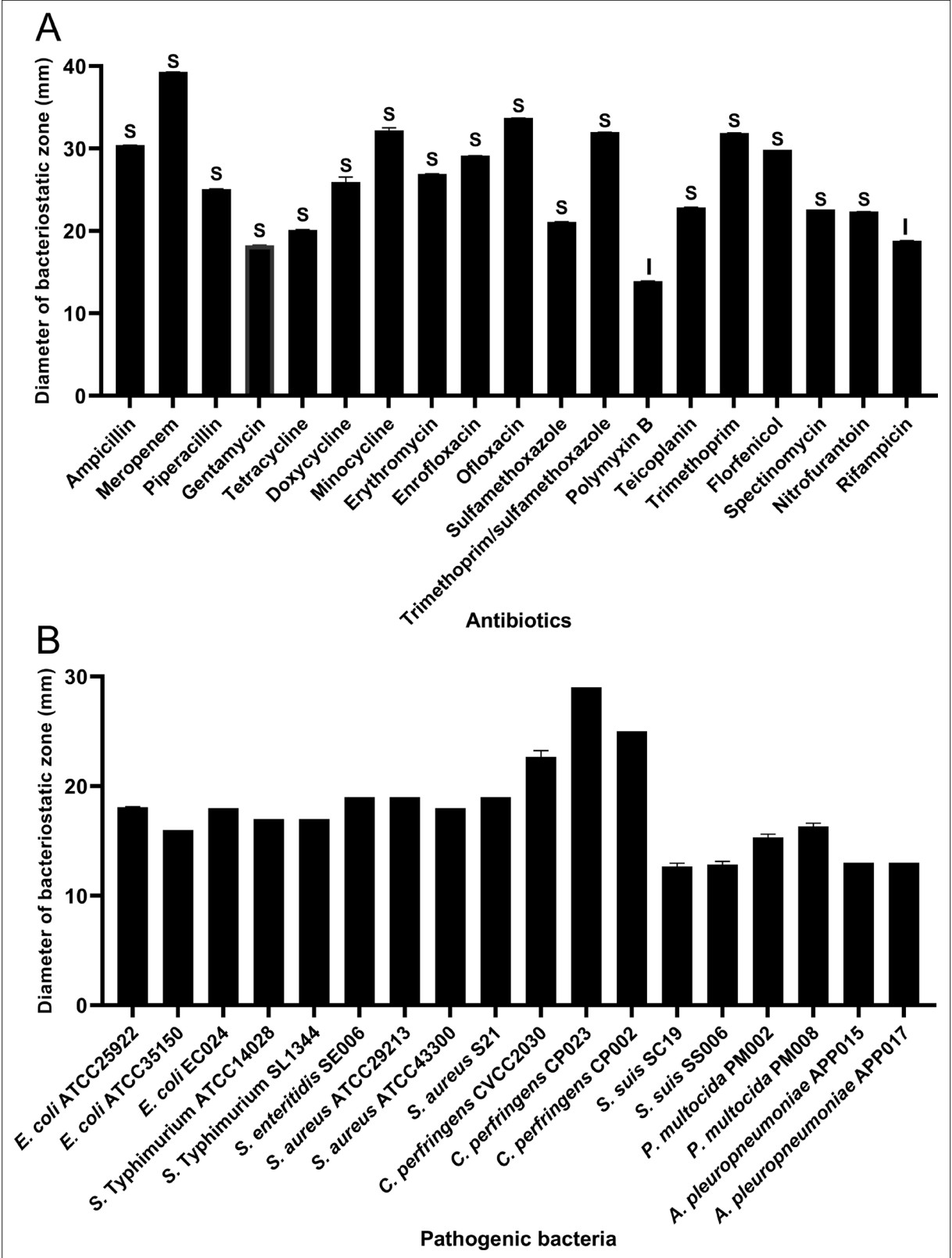

**Figure 2.** Antibiotic susceptibility of *B. velezensis* HBXN2020 and bacteriostasis assay in vitro. (**A**) The diameter of the antibacterial zone indicates the extent of sensitivity to antibiotics. (**B**) The diameter of the antibacterial zone indicates the extent of inhibition against pathogenic bacteria. The diameter of the antibacterial zone was measured with vernier caliper. Each group was repeated three times (n=3). R, resistant; I, moderately sensitive; S, sensitive.

*Figure 2 continued on next page*

*Figure 2 continued*

The online version of this article includes the following source data and figure supplement(s) for figure 2:

**Source data 1.** Raw data values for *Figure 2A and B*.

**Figure supplement 1.** In vitro antibacterial test of *B. velezensis* HBXN2020.

**Figure supplement 1—source data 1.** PDF file containing original *B. velezensis* HBXN2020 cell-free supernatant (CFS) antibacterial images for *Figure 2—figure supplement 1*.

**Figure supplement 1—source data 2.** Original files for *B. velezensis* HBXN2020 cell-free supernatant (CFS) antibacterial images displayed in *Figure 2—figure supplement 1*.

evaluate the antibacterial activity of HBXN2020. The results showed that HBXN2020 had a similar inhibitory effect on standard and wild strains of *E. coli*, *Salmonella*, *S. aureus*, and *Clostridium perfringens* (*C. perfringens*), as well as wild strains of *Streptococcus suis* (*S. suis*), *Pasteurella multocida* (*P. multocida*), and *Actinobacillus pleuropneumoniae* (*A. pleuropneumoniae*). Except for the wild strains of *S. suis* and *A. pleuropneumoniae*, the diameter of the inhibition zone of other strains was above 15 mm (*Figure 2B*, *Figure 2—figure supplement 1*, and *Figure 2—source data 1*), while the size of the inhibition zone of *S. suis* and *A. pleuropneumoniae* was also more than 12 mm.

## Genomic characteristics

HBXN2020's complete genome was sequenced using the Illumina HiSeq and PacBio platforms. The results showed that HBXN2020 has a circular chromosome of 3,929,792 bp (*Figure 3A*) with a GC content of 46.5%, including 3744 protein-coding genes (coding sequence [CDS]), 86 tRNA genes, and 27 rRNA genes (*Supplementary file 2*). Furthermore, the HBXN2020 genome also contains 13 different clusters of secondary metabolic synthesis genes, such as Fengycin (genomic position: 1,865,856) and Difficidin (genomic position: 2,270,091) (*Supplementary file 2*).

A phylogenetic tree based on genome-wide data from all 14 *Bacillus* strains demonstrated that the HBXN2020 belongs to the *B. velezensis* species (*Figure 3B*). To further understand the classification status of HBXN2020, the online tool JSpeciesWS was used to calculate the average nucleotide

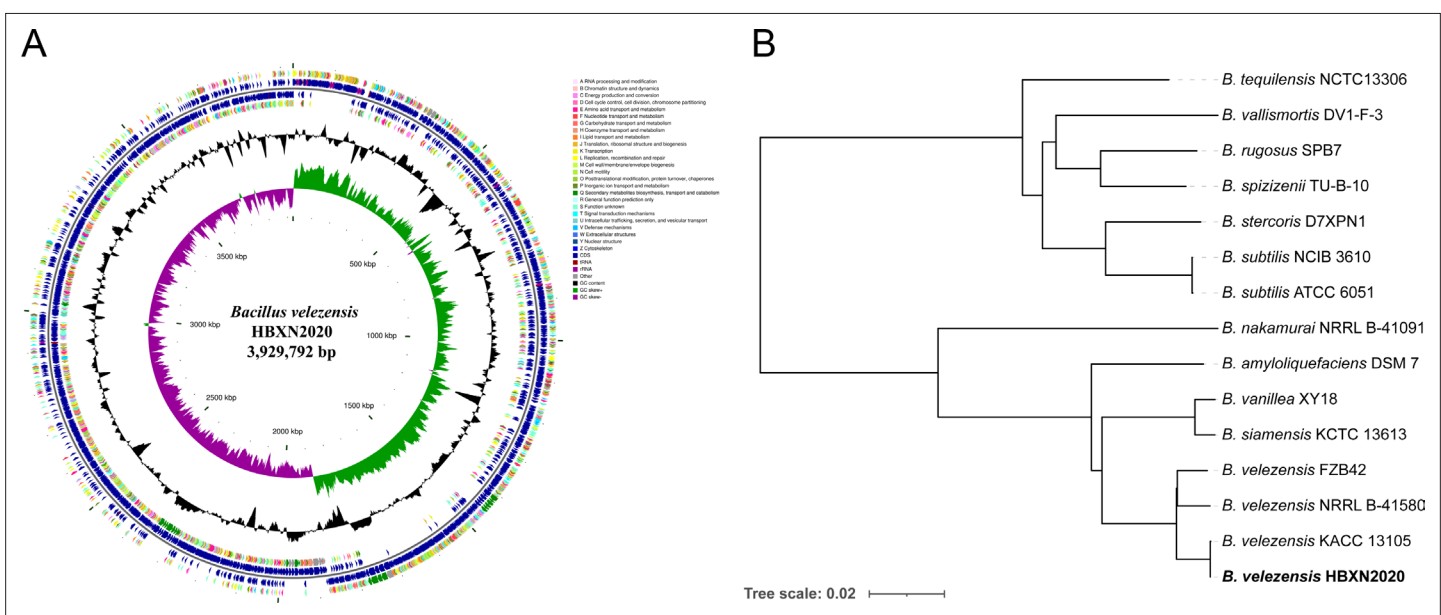

**Figure 3.** Genomic characteristics and phylogenetic relationships of *B. velezensis* HBXN2020. (**A**) The whole-genome map of *B. velezensis* HBXN2020 with its genomic features. The map consists of six circles. From the inner circle to the outer circle: (1) GC-skew, (2) GC content, (3) reverse protein-coding genes, different colors represent different COG functional classifications, (4) genes transcribed in reverse direction, (5) genes transcribed in forward direction, (6) forward protein-coding genes, different colors represent different COG functional classifications. (**B**) The whole-genome phylogenetic tree was constructed based on genome-wide data from 14 *Bacillus* strains. *B. velezensis* HBXN2020 are indicated in bold.

The online version of this article includes the following figure supplement(s) for figure 3:

**Figure supplement 1.** *B. velezensis* HBXN2020 genome average nucleotide identity (ANI) analysis.

identity (ANI) based on the BLAST (ANIb) method. As shown in *Figure 3—figure supplement 1*, HBXN2020 was found to be a member of the *B. velezensis* species due to the high percentage of ANIb (more than 97%).

## Safety evaluation of *B. velezensis* HBXN2020

We assessed the in vivo effects of *B. velezensis* HBXN2020 in a mouse model in order to ascertain its safety. After gavage with *B. velezensis* HBXN2020 spores for 2 weeks, we observed no significant difference in the body weight of each group of mice (*Figure 4A*, *Figure 4—source data 1*). The gene expression levels of *Tnfa*, *Il1b*, *Il6*, and *Il10* of colon from all groups of mice exhibited no remarkable changes (*Figure 4B*, *Figure 4—source data 1*). However, in the colon, mRNA levels of the barrier proteins *Tjp1* and *Ocln* were trending toward an increase compared with the control group (*Figure 4C*, *Figure 4—source data 1*). Additionally, blood routine tests and serum biochemistry tests were performed for mice in the control group and H-HBXN2020 group on the 14th day after oral administration of *B. velezensis* HBXN2020 spore multiple times. As shown in *Figure 4D*, *Figure 4—source data 1*, the blood parameters of mice after *B. velezensis* HBXN2020 spores treatment, including red blood cells (RBC), mean corpuscular volume (MCV), mean corpuscular hemoglobin (MCH), hemoglobin (HGB), white blood cells (WBC), platelets (PLT), hematocrit (HCT), and mean corpuscular hemoglobin concentration (MCHC), were consistent with those of healthy mice (*Figure 4—figure supplement 1*, *Figure 4—figure supplement 1—source data 1*). The serum biochemical parameters, including alanine aminotransferase (ALT), aspartate aminotransferase (AST), albumin (ALB), total bilirubin (TBIL), serum creatinine (CREA), and blood urea nitrogen (BUN), were also within normal limits (*Figure 4E*, *Figure 4—source data 1*). The corresponding histological analysis of colon tissue from mice receiving low, medium, and high doses of *B. velezensis* HBXN2020 spores (L-HBXN2020, M-HBXN2020, and H-HBXN2020 groups, respectively) is presented in *Figure 4F*. The colon tissue sections of mice in the test groups showed no significant differences compared to the control group. Additionally, there were no observable differences in the major organ tissues (heart, liver, spleen, lung, and kidney) of mice treated with the high dose of *B. velezensis* HBXN2020 spores compared to healthy mice (*Figure 4—figure supplement 2*). The results of this trial show that *B. velezensis* HBXN2020 is safe to use and has no negative side effects in mice.

## Oral administration of *B. velezensis* HBXN2020 spores alleviated infection by *S.* Typhimurium

Testing both solid and liquid co-culture methods to see if *B. velezensis* HBXN2020 might suppress *S.* Typhimurium ATCC14028 (STm), the findings indicated that *B. velezensis* HBXN2020 decreased the development of STm in a dose-dependent manner (*Figure 5A*, *Figure 5—figure supplement 1*, *Figure 5—source data 1*, and *Figure 5—figure supplement 1—source data 1*). Next, the therapeutic potential of *B. velezensis* HBXN2020 was evaluated in an STm-infected mouse model. At days 1, 3, and 5 after STm infection, mice in the STm+HBXN2020 group and HBXN2020 group were orally administered *B. velezensis* HBXN2020 spores by gavage, STm+CIP group were given oral ciprofloxacin, while the control group and STm+PBS group were orally treated with sterile PBS (*Figure 5B*). Also, the number of excreted STm in feces was counted daily at the designated time points. As shown in *Figure 5C*, *Figure 5—source data 1*, there was a significant and continuous reduction in the number of STm in feces following treatment with *B. velezensis* HBXN2020 spores or ciprofloxacin, while the number of *B. velezensis* HBXN2020 viable bacteria in feces is also gradually decreasing (*Figure 5—figure supplement 2*, *Figure 5—figure supplement 2—source data 1*). The number of STm in the mice feces of STm+HBXN2020 group and STm+CIP group exhibited a reduction ranging from 0.12 to 1.18 logs and 0.76–1.81 logs, respectively, compared with the STm+PBS group, over the period from day 2 to day 7 post-treatment. Moreover, treatment with *B. velezensis* HBXN2020 spore or ciprofloxacin resulted in a 1.06, 1.69, 1.14 logs and 1.70, 2.15, 1.48 logs reduction in the number of STm in the cecum, colon, and ileum, respectively (*Figure 5D and E*, *Figure 5—figure supplement 2*, *Figure 5—source data 1*, and *Figure 5—figure supplement 2—source data 1*). The weight loss (p<0.05) and disease activity index (DAI) scores (p<0.01) were significantly reduced in mice after *B. velezensis* HBXN2020 spore treatment or ciprofloxacin treatment compared to that in the PBS treatment group (*Figure 5F and G*, *Figure 5—source data 1*). In addition, we also measured the colon length of the mice and found that the STm+PBS group had a significantly shorter colon length

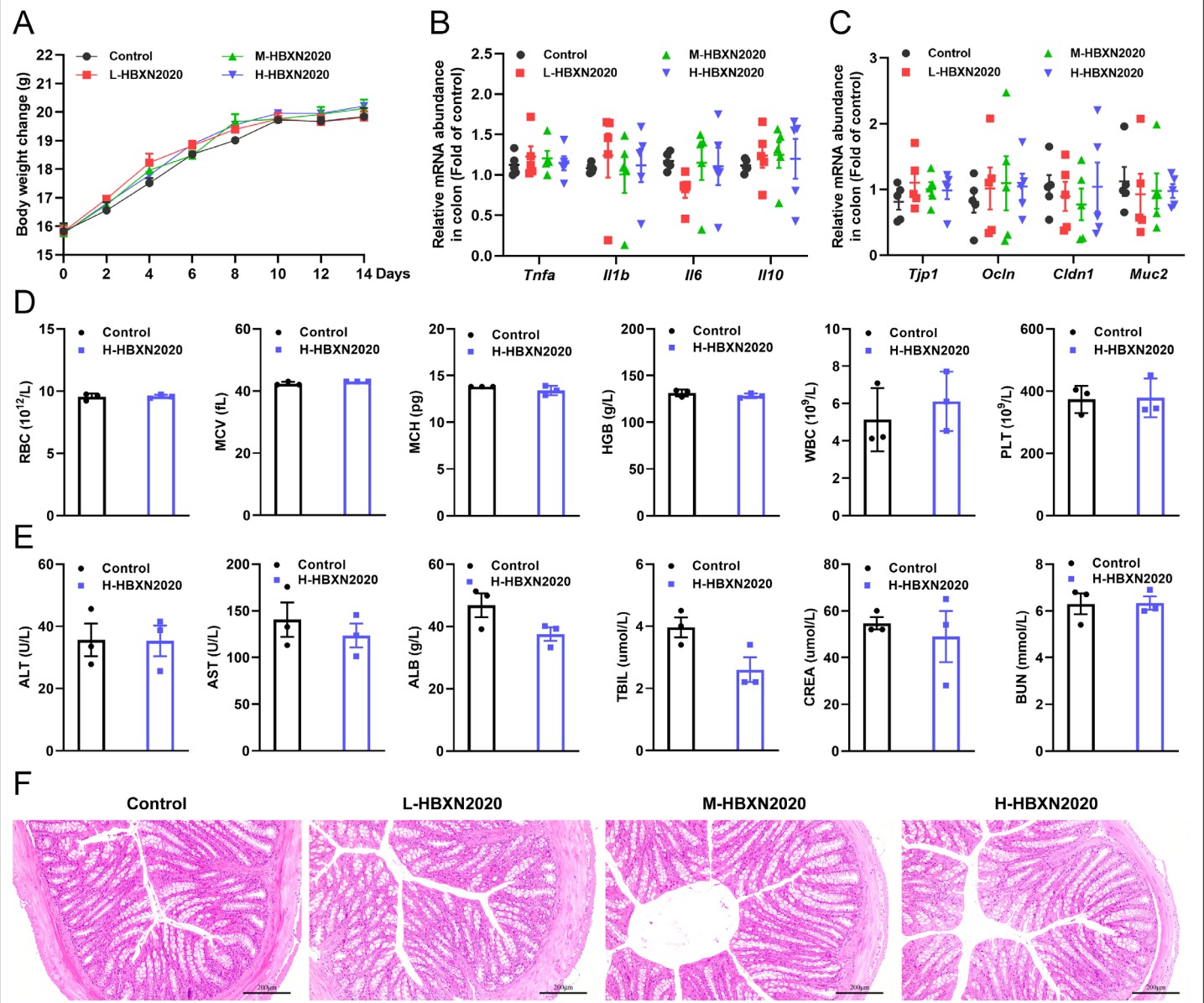

**Figure 4.** In vivo safety evaluation of *B. velezensis* HBXN2020 in a mouse model. (**A**) Body weights changes of mice during gavage with *B. velezensis* HBXN2020 spores. Mice were treated with sterile PBS (control group) or low-dose (L-HBXN2020 group), medium dose (M-HBXN2020 group), and high-dose (H-HBXN2020 group) of *B. velezensis* HBXN2020 spores. Weighing and gavage were performed once every 2 days during the experimental period (15 days). Data were shown as mean values ± SEM (n=5). (**B**) The mRNA levels of inflammatory cytokines in the colon of mice measured by RT-qPCR. Data were shown as mean values ± SEM (n=5). (**C**) The mRNA levels of barrier protein *Tjp1*, *Ocln*, *Cldn1*, and *Muc2* in the colon of mice measured by RT-qPCR. Data were shown as mean values ± SEM (n=5). (**D**) Major blood routine parameters and (**E**) serum biochemical parameters of mice in the control group and H-HBXN2020 group. Data were shown as mean values ± SEM (n=3). (**F**) Hematoxylin and eosin (H&E) stained colon sections in the different groups. Scale bar: 200 μm.

The online version of this article includes the following source data and figure supplement(s) for figure 4:

**Source data 1.** Raw data values for *Figure 4A–E*.

**Figure supplement 1.** Hematological parameters.

**Figure supplement 1—source data 1.** Raw data values for *Figure 4—figure supplement 1A and B*.

**Figure supplement 2.** Hematoxylin and eosin (H&E) staining of the intestine, heart, liver, spleen, lung, and kidney.

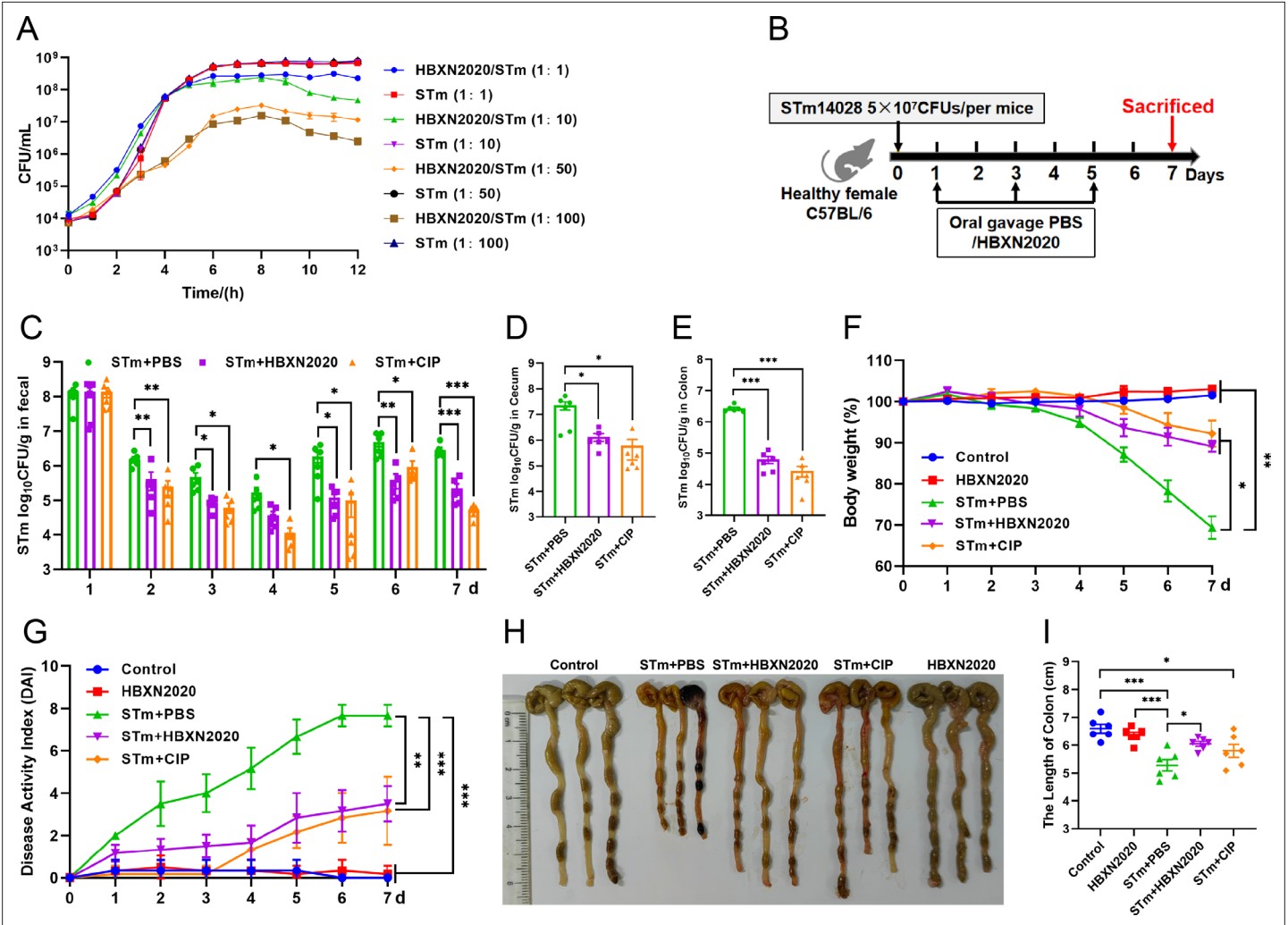

**Figure 5.** Oral *B. velezensis* HBXN2020 spores alleviated infection by *S*. Typhimurium. (**A**) In vitro bacterial competition between STm and *B. velezensis* HBXN2020. STm were co-incubated with *B. velezensis* HBXN2020 at various ratios at 37°C with shaking. The growth of STm was reflected by bacterial counting per hour. (**B**) Experimental design for treatment in this study. Orally administrated with either sterile PBS, *B. velezensis* HBXN2020 spores or ciprofloxacin by gavage at days 1, 3, and 5 after STm ($5×10^7$ CFU/mouse) infection, respectively. All mice were euthanized at day 7 after STm infection. (**C**) Bacterial count of STm in mouse feces. Fecal samples were collected per day after STm infection and resuspended in sterile PBS (0.1 g of fecal resuspended in 1 mL of sterile PBS). One hundred microliters of each sample performed a serial of 10-fold dilutions and spread on selective agar plates (50 µg/mL kanamycin) and incubated at 37°C for 12 hr before bacterial counting. The bacterial loads of STm in (**D**) cecum and (**E**) colon. The cecum and colon were harvested and then homogenized. Data were shown as mean values ± SEM (n=6). Statistical significance was evaluated using Student's t-test (*, $p<0.05$, **, $p<0.01$, and ***, $p<0.001$). (**F**) Daily body weight changes and (**G**) daily disease activity index (DAI) scores of mice with different treatment groups. Data were shown as mean values ± SEM (n=6). Statistical significance was evaluated using one-way analysis of variance (ANOVA) with Tukey's multiple comparisons test (*, $p<0.05$, **, $p<0.01$, and ***, $p<0.001$). (**H**) Colonic tissue images. (**I**) The length of the colon from per group (n=6). Statistical significance was evaluated using one-way ANOVA with Tukey's multiple comparisons test (*, $p<0.05$, **, $p<0.01$, and ***, $p<0.001$).

The online version of this article includes the following source data and figure supplement(s) for figure 5:

**Source data 1.** Raw data values for *Figure 5A, C, D, E, F, G, I*.

**Source data 2.** PDF file containing original colonic tissue image for *Figure 5H*.

**Source data 3.** Original files for colonic tissue image displayed in *Figure 5H*.

**Figure supplement 1.** In vitro antagonistic activity of *B. velezensis* HBXN2020 against *S*. Typhimurium ATCC14028 in liquid and solid media.

**Figure supplement 1—source data 1.** Raw data values for *Figure 5—figure supplement 1C and D*.

**Figure supplement 1—source data 2.** PDF file containing original *B. velezensis* HBXN2020 and its cell-free supernatant (CFS) inhibited STm images for *Figure 5—figure supplement 1 and B*.

**Figure supplement 1—source data 3.** Original files for antibacterial images of *B. velezensis* HBXN2020 and its cell-free supernatant (CFS) against STm

*Figure 5 continued on next page*

*Figure 5 continued*

displayed in *Figure 5—figure supplement 1A and B*.

**Figure supplement 2.** The effect of therapeutic *B. velezensis* HBXN2020 on the bacterial load in feces of mice and the histological score of the colons.

**Figure supplement 2—source data 1.** Raw data values for *Figure 5—figure supplement 2A–C*.

(p<0.001) than the control group and HBXN2020 group. However, compared to the STm+PBS group, the STm+HBXN2020 group (p<0.05) and STm+CIP group exhibited a longer colon (*Figure 5H and I*, *Figure 5—source data 1*).

Histological analysis further showed that in the control group and HBXN2020 group, the colon epithelial cells and crypt structure were intact with neat villi, while STm+PBS group showed significant histological damage, including erosion or loss of the intestinal epithelium, crypt destruction, and inflammatory cell infiltration in the colonic tissues (*Figure 6A*). Compared with the STm+PBS group, *B.*

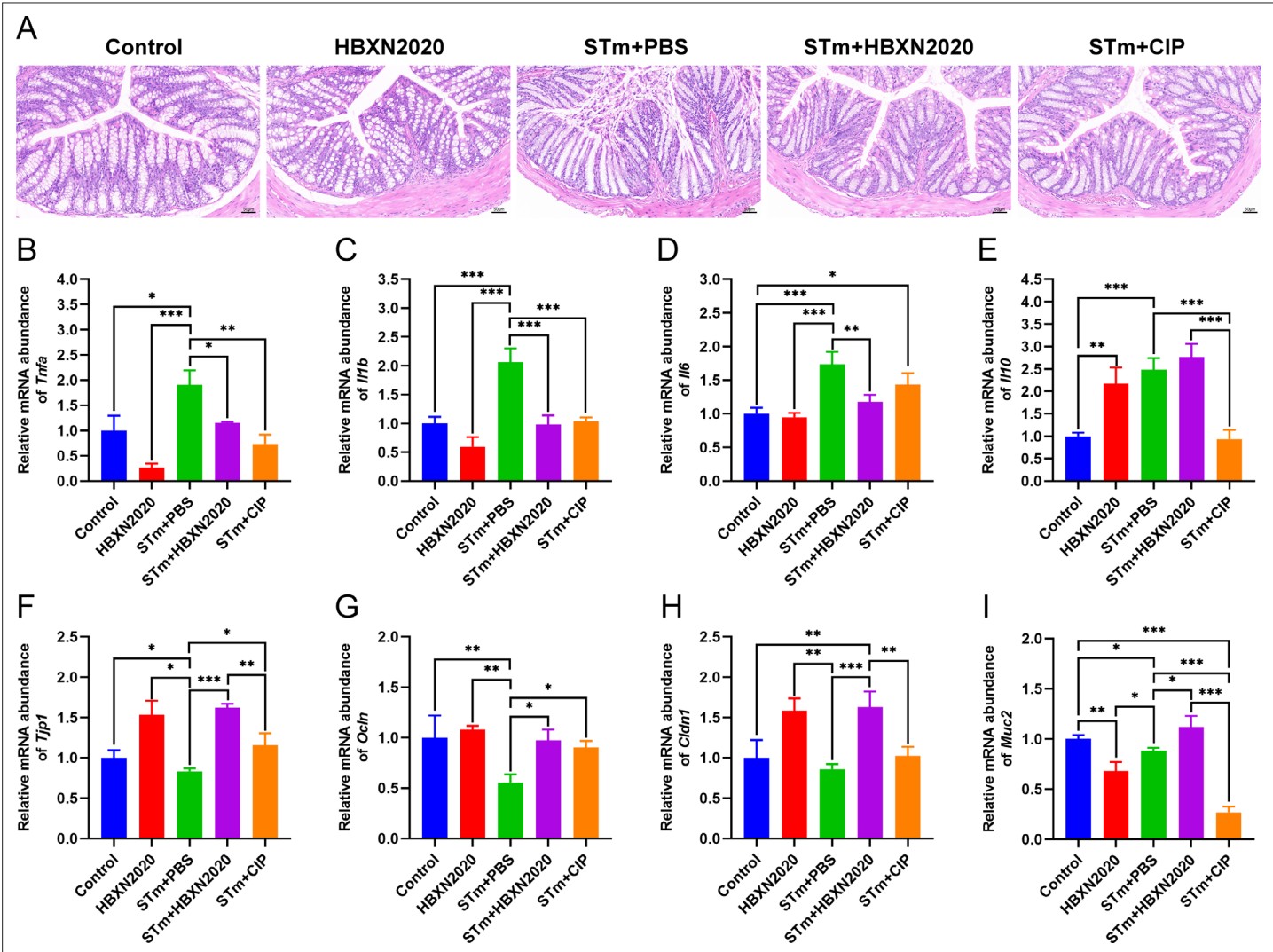

**Figure 6.** Oral *B. velezensis* HBXN2020 spores attenuated colonic damage and inflammatory reaction. (**A**) Hematoxylin and eosin (H&E) stained colon tissue sections. Scale bar: 50 μm. The mRNA levels of (**B**) *Tnfa*, (**C**) *Il1b*, (**D**) *Il6*, and (**E**) *Il10* were detected by RT-qPCR. Data were shown as mean values ± SEM (n=5). The mRNA levels of (**F**) *Tjp1*, (**G**) *Ocln*, (**H**) *Cldn1*, and (**I**) *Muc2* in colon tissue were detected by RT-qPCR. Data were shown as mean values ± SEM (n=5). Statistical significance was evaluated using one-way analysis of variance (ANOVA) with Tukey's multiple comparisons test (*, p<0.05, **, p<0.01, and ***, p<0.001).

The online version of this article includes the following source data for figure 6:

**Source data 1.** Raw data values for *Figure 6B–I*.

*velezensis* HBXN2020 and ciprofloxacin protected the mucosal architecture and the loss of intestinal epithelial cells, and reduced inflammatory cell infiltration (*Figure 6A*).

To further assess the impact of *B. velezensis* HBXN2020 on intestinal inflammatory response, the mRNA levels of inflammatory cytokines in the colonic tissues were measured. As shown in *Figure 6B–D*; *Figure 6—source data 1*, the mRNA levels of *Tnfa* (p<0.05), *Il1b* (p<0.001), *Il6* (p<0.001), and *Il10* (p<0.001) in the colon tissue were significantly increased in the STm+PBS group compared to the control group. *B. velezensis* HBXN2020 spore treatment significantly decreased the mRNA levels of *Tnfa* (p<0.05), *Il1b* (p<0.001), *Il6* (p<0.01), and increased the *Il10* level, compared with the STm+PBS group. In contrast, ciprofloxacin treatment failed to significantly decrease the mRNA levels of *Il6* in the colon of STm-infected mice (*Figure 6D*, *Figure 6—source data 1*). Similarly, ciprofloxacin treatment drastically dampened the mRNA levels of *Il10* in the colon of STm-infected mice (*Figure 6E*, *Figure 6—source data 1*).

To assess the influence of *B. velezensis* HBXN2020 on the intestinal barrier integrity, the mRNA levels of tight junction proteins in the colonic tissues were measured. As presented in *Figure 6*, compared with control group, the mRNA levels of tight junction protein in the STm+PBS group was significantly reduced, including *Tjp1* (p<0.05), *Ocln* (p<0.01), *Cldn1*, and *Muc2* (p<0.05) (*Figure 6F–I*, *Figure 6—source data 1*). Moreover, the levels of tight junction protein transcription were remarkably elevated in the STm+HBXN2020, STm+CIP, and HBXN2020 group, including *Tjp1* (p<0.001, p<0.05, and p<0.05, respectively), *Ocln* (p<0.05, p<0.05, and p<0.01, respectively), and *Cldn1* (p<0.001, p=0.0857, and p<0.01, respectively), except for *Cldn1* (p=0.0857) in group STm+CIP, compared with that of the STm+PBS group (*Figure 6F–I*, *Figure 6—source data 1*). In addition, we also detected the transcription level of *Muc2* in colon tissue, which mainly exists in and around goblet cells and is an important component of the mucus layer. The transcription levels of *Muc2* were dramatically increased in the STm+HBXN2020 group compared with the STm+PBS group (*Figure 6I*, *Figure 6—source data 1*). Conversely, the mRNA levels of *Muc2* were significantly decreased in the HBXN2020 group and STm+CIP group (*Figure 6I*, *Figure 6—source data 1*).

Next, we further explore the impact of *B. velezensis* HBXN2020 on the intestinal microbiota composition of STm-treated mice by 16S rRNA gene high-throughout sequencing. As shown in *Figure 7*, alpha diversity analysis revealed that both the richness and diversity (calculated in Sobs, Chao, Shannon, and Simpson indexes) were lower in STm+PBS group than control group, HBXN2020 group, STm+HBXN2020 group, and STm+CIP group (*Figure 7A–D*, *Figure 7—source data 1*). Although an increasing trend of alpha diversity was observed in STm+CIP group, but without statistical difference was achieved as compared with other groups. Principal components analysis (PCA) based on Bray-Curtis distance showed that a separation in the gut microbiota structure among control and STm+PBS group. When *B. velezensis* HBXN2020 spores were supplemented, the gut microbiota composition of the STm+HBXN2020 group, HBXN2020 group, and the control group were closer together ($R^2$=0.1616, p=0.006, *Figure 7E*).

We analyzed the community composition of colonic microbiota at the phylum and genus level, and the results revealed that the composition of the gut microbiota changed markedly after STm-infected mice. At the phylum level, *Bacteroidetes*, *Firmicutes*, *Verrucomicrobiota*, and *Proteobacteria* were predominant phyla in the fecal microbiota (*Figure 7F*, *Figure 7—figure supplement 1*, and *Figure 7—source data 1*). At the genus level, infection with STm in the STm+PBS group dramatically reduced the relative abundance of *norank_f_Muribaculaceae*, *Lactobacillus*, and *Akkermansia* (*Figure 7G and H–J*, *Figure 7—source data 1*) and enhanced the abundance of *Enterococcus*, *Salmonella*, *Bacteroides*, *Alloprevotella*, and *Escherichia-Shigella* compared to the control group (*Figure 7G and K*, *Figure 7— figure supplement 1*, and *Figure 7—source data 1*). In contrast, the STm+HBXN2020 group and HBXN2020 group significantly improved the relative abundance of *norank_f_Muribaculaceae* (p=0.49, p<0.05) and *Lactobacillus* (p<0.05, p<0.001) compared with the STm+PBS group (*Figure 7H and I*, *Figure 7—source data 1*). However, ciprofloxacin treatment significantly decreased the relative abundance of *Lactobacillus* (p<0.05) and *Akkermansia* (p<0.05) compared with the STm+HBXN2020 group (*Figure 7I and J*, *Figure 7—source data 1*). Differentially abundant fecal bacterial taxa in STm-treated mice in response to *B. velezensis* HBXN2020 and ciprofloxacin were identified by linear discriminant analysis (LDA) effect size (LEfSe) analysis (*Figure 7L*). Additionally, we found that two bacterial genera including *Enterococcus* and *Streptococcus* were enriched in the STm+PBS group, while the other three taxa were enriched including *Akkermansia*, *Family_XIII_AD3011_group*, and *Corynebacterium* in

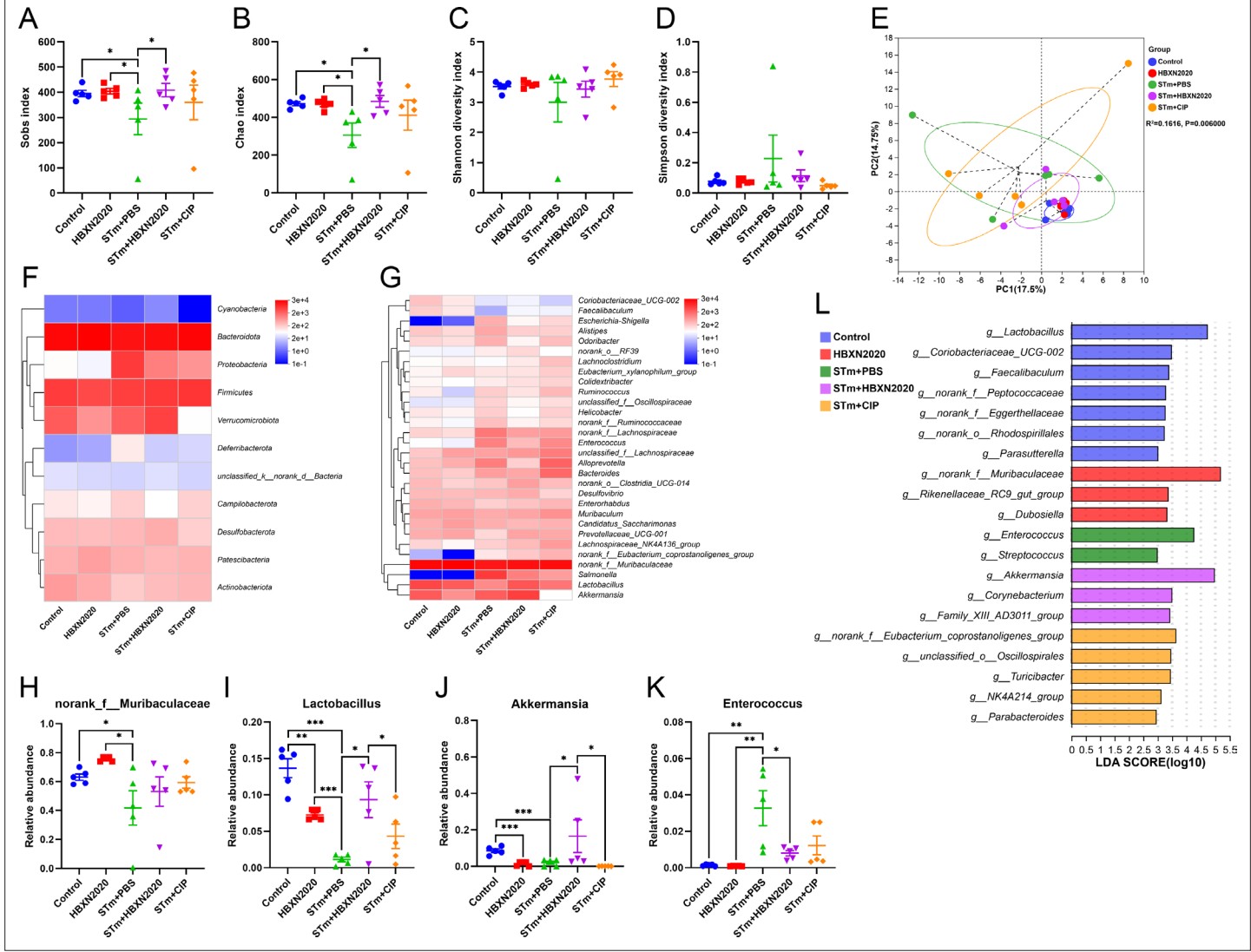

**Figure 7.** Oral *B. velezensis* HBXN2020 spores regulated the composition of intestinal microbiota. The alpha diversity of the gut microbiota, determined by the (**A**) Sobs, (**B**) Chao, (**C**) Shannon, and (**D**) Simpson index. Data were shown as mean values ± SEM (n=5). (**E**) The principal components analysis (PCA) plot showed the β diversity of the gut microbiota based on Bray-Curtis distance at the operational taxonomic unit (OTU) level. Heatmap of the community composition of colonic microbiota at the phylum (top 15 phyla) (**F**) and genus (top 30 genera) (**G**) level. Relative abundance of selected taxa (**H**) *norank_f_Muribaculaceae*, (**I**) *Lactobacillus*, (**J**) *Akkermansia*, and (**K**) *Enterococcus*. Data were shown as mean values ± SEM (n=5). (**L**) Analysis of differences in the microbial communities by LEfSe (linear discriminant analysis [LDA] effect size) (LDA score>2) among different groups. Significance was evaluated by one-way analysis of variance (ANOVA) with Tukey's multiple comparisons test (*, $p<0.05$, **, $p<0.01$, and ***, $p<0.001$).

The online version of this article includes the following source data and figure supplement(s) for figure 7:

**Source data 1.** Raw data values for *Figure 7A–D and H–K*.

**Figure supplement 1.** Analysis of the community compositions of colonic microbiota.

the STm+HBXN2020 group, *Parabacteroides*, *NK4A214_group*, *Turicibacter*, *unclassified_o__Oscillo-spirales*, and *norank_f__Eubacterium_coprostanoligenes_group* in the STm+CIP group, and *norank_f_Muribaculaceae*, *Rikenellaceae_RC9_gut_group* and *Dubosiella* in the HBXN2020 group (*Figure 7L*).

## Prophylactic *B. velezensis* HBXN2020 spores alleviated infection by *S.* Typhimurium

Based on the improved therapeutic efficacy of *B. velezensis* HBXN2020 in the treatment of STm-infected mice, we further explored its potential for disease prevention by evaluating pretreatment. Mice in the PBS+STm group, HBXN2020+STm group, and CIP+STm group were given the same

amount of sterile PBS, *B. velezensis* HBXN2020 spores or ciprofloxacin by gavage 1 week in advance. As shown in *Figure 8*, compared with the PBS+STm group, oral administration of *B. velezensis* HBXN2020 spores or ciprofloxacin as a preventive measure remarkably alleviated infection by STm, including weight loss of mice (p<0.01) (*Figure 8B*, *Figure 8—source data 1*), a significant reduction in DAI (p<0.001) (the comprehensive score of weight loss, stool consistency, and blood in the feces, *Figure 8C*, *Figure 8—source data 1*), and the prevention of colon length shortening (*Figure 8D and E*, *Figure 8—source data 1*). Meanwhile, the number of *B. velezensis* HBXN2020 viable bacteria in feces is also gradually decreasing with time prolonging (*Figure 8—figure supplement 1*, *Figure 8— figure supplement 1—source data 1*). Furthermore, compared with the PBS+STm group, oral administration of *B. velezensis* HBXN2020 spores or ciprofloxacin not only reduced the number of STm in mouse feces (*Figure 8F*, *Figure 8—source data 1*) but also decreased STm colonization in the ileum, cecum, and colon (*Figure 8G and H*, *Figure 8—figure supplement 1*, *Figure 8—source data 1*, and *Figure 8—figure supplement 1—source data 1*).

Histological analysis further revealed that prophylactic *B. velezensis* HBXN2020 spore suppressed STm-induced loss of the intestinal epithelium, inflammatory cell infiltration, and colonic mucosa damage (*Figure 9A*), while prophylactic ciprofloxacin failed to inhibit the loss of colonic epithelial cells. We next measured the mRNA levels of inflammatory cytokines and tight junction proteins in colon tissue in the prophylactic animal experiment. As with the treatment test, prophylactic *B. velezensis* HBXN2020 spores largely attenuated the mRNA levels of *Tnfa* (p<0.001), *Il1b* (p<0.001), and *Il6* (p<0.001), and significantly increased levels of *Il10* (p<0.05), in the colon tissue of mice compared with the PBS+STm group (*Figure 9B–E*, *Figure 9—source data 1*). Similarly, prophylactic ciprofloxacin significantly reduced the mRNA levels of *Tnfa* (p<0.01) and *Il1b* (p<0.001), and drastically elevated levels of *Il10* (p<0.001) in the colon tissue of mice compared with the PBS+STm group (*Figure 9B, C, and E*, *Figure 9—source data 1*). Also, prophylactic *B. velezensis* HBXN2020 spores significantly increased the transcription levels of *Tjp1* (p<0.001), *Ocln* (p<0.001), *Cldn1* (p<0.05), and *Muc2* (p<0.001) compared with that of the PBS+STm group (*Figure 9F–I*, *Figure 9—source data 1*). Notably, the mRNA levels of *Tjp1* (p<0.001), *Ocln* (p<0.05), and *Muc2* (p<0.05) in the CIP+STm group were significantly lower than those in the HBXN2020+STm group. Overall, these results suggested that prophylactic *B. velezensis* HBXN2020 spores was capable of alleviating STm-induced colonic injury and inflammation.

Next, we examined the impact of prophylactic *B. velezensis* HBXN2020 on the intestinal microbiota composition of STm-treated mice. Sobs, Chao, and Shannon index were significantly increased in the HBXN2020, PBS+STm group, HBXN2020+STm group, and CIP+STm group compared with the control group (*Figure 10A–C*, *Figure 10—source data 1*), while the Simpson index (not statistically different) significantly decreased (*Figure 10D*, *Figure 10—source data 1*). PCA showed significant separation between control and PBS+STm groups (R²=0.3186, p=0.014, *Figure 10E*). When *B. velezensis* HBXN2020 spores or ciprofloxacin were supplemented, the community clustering was significantly similar to that of the control group rather than the PBS+STm group (*Figure 10E*). Meanwhile, the overall fecal bacterial composition of mice at the phylum and genus level in all groups was similar to that of the treatment experiment. The fecal microbiota was dominated by phyla *Firmicutes*, *Bacteroidetes*, *Verrucomicrobiota*, *Patescibacteria*, and *Actinobacteria* in all five groups (*Figure 10— figure supplement 1*). At the genus level, PBS+STm group exhibited lower relative abundances of *Lactobacillus* and *Akkermansia* (*Figure 10G–I*, *Figure 10—source data 1*), and higher relative abundances of *Alistipes*, *Bacteroides*, *Escherichia-Shigella*, *Enterococcus*, and *Alloprevotella*, compared with the control (*Figure 10G and J–M*, *Figure 10—figure supplement 1*, and *Figure 10—source data 1*). Moreover, compared with the PBS+STm group, following supplementation with *B. velezensis* HBXN2020 spores or ciprofloxacin, *Lactobacillus* significantly increased, and the gut microbiota was restored to a composition similar to the control group. It was noteworthy that six bacterial genera including *Alistipes*, *Odoribacter*, and *norank_f__Oscillospiraceae* were enriched in PBS+STm group, while other three taxa (e.g. *Lactobacillus* and *Akkermansia*) were enriched in the control group (*Figure 10N*). Moreover, we found that *Candidatus_Arthromitus* was enriched in the HBXN2020+STm group, *Dubosiella* was enriched in the CIP+STm group, and *Faecalibaculum*, *Rikenellaceae_RC9_gut_ group*, and *unclassified_f__Lactobacillaceae* were enriched in the HBXN2020 group (*Figure 10N*). Above all, these results indicated that prophylactic *B. velezensis* HBXN2020 spores regulated the gut microbiota composition, leading to the potential to attenuate the STm-induced dysbiosis.

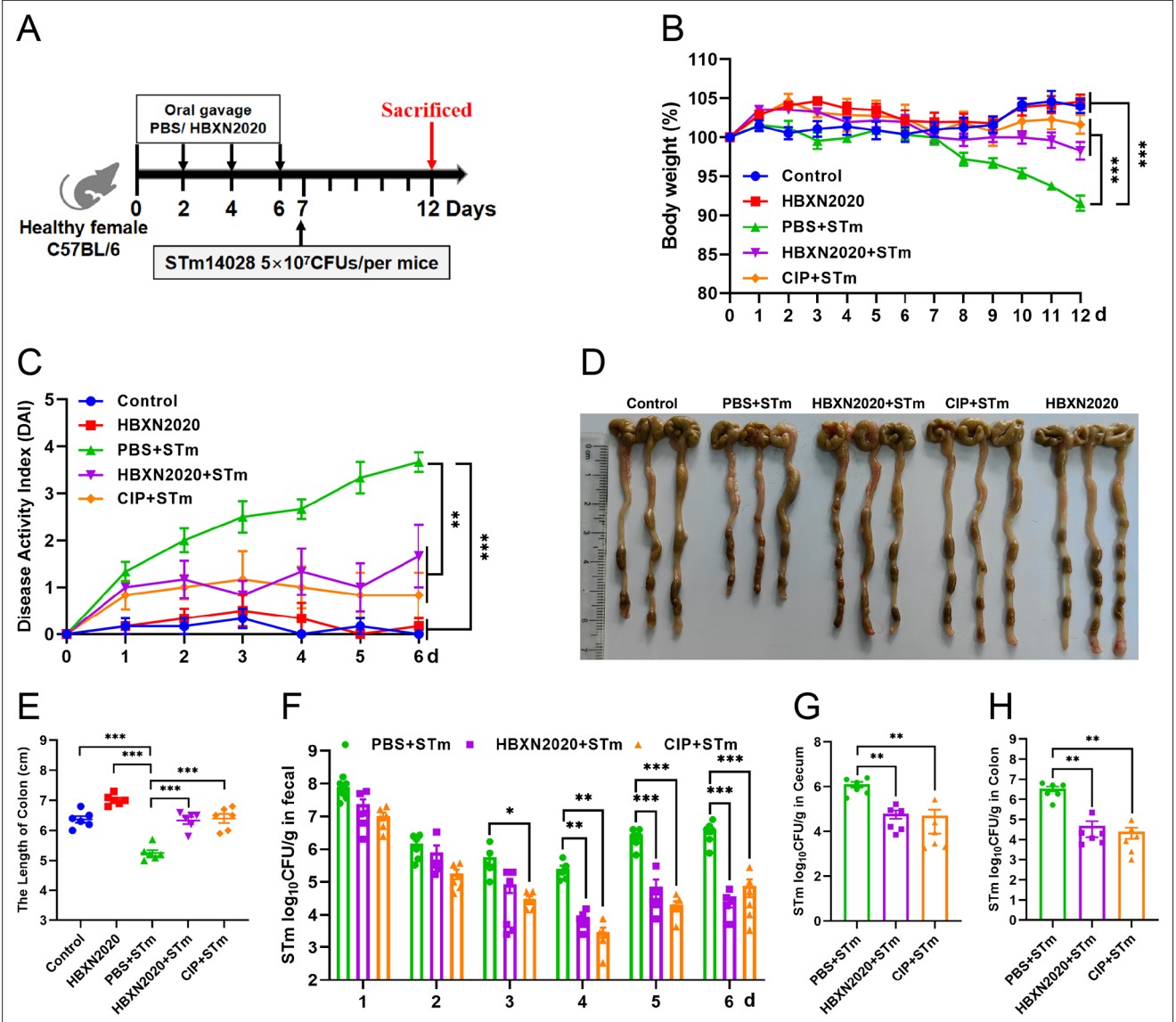

**Figure 8.** Prophylactic *B. velezensis* HBXN2020 spores attenuated the symptoms of *S.* Typhimurium-infected mouse. (**A**) Experimental design for treatment in this study. At days 1, 3, 5, and 7, each mouse in the HBXN2020+STm group, CIP+STm group, and PBS+STm group were received 200 μL of *B. velezensis* HBXN2020 spores (1×10⁸ CFU/mouse), ciprofloxacin or sterile PBS by gavage, respectively. Then, mice in PBS+STm group, HBXN2020+STm group, and CIP+STm group were orally inoculated with 200 μL (5×10⁷ CFU/mouse) of STm on day 7, respectively. On day 12, all mice were euthanized. (**B**) Daily body weight changes and (**C**) daily disease activity index (DAI) scores of mice with different groups following STm treatment. Data were shown as mean values ± SEM (n=6). Statistical significance was evaluated using one-way analysis of variance (ANOVA) with Tukey's multiple comparisons test (*, p<0.05, **, p<0.01, and ***, p<0.001). (**D**) Colonic tissue images. (**E**) The length of the colon from per group (n=6). (**F**) Bacterial count of STm in mouse feces. Fecal samples were collected every day after STm infection and resuspended in sterile PBS (0.1 g of fecal resuspended in 1 mL of sterile PBS) (n=6). One hundred microliters of each sample performed a serial of 10-fold dilutions and spread on selective agar plates (50 μg/mL kanamycin) and incubated at 37°C for 12 hr before bacterial counting. The bacterial loads of STm in (**G**) cecum and (**H**) colon (n=6). The cecum and colon were harvested and then homogenized. Statistical significance was evaluated using Student's t-test (*, p<0.05, **, p<0.01, and ***, p<0.001).

The online version of this article includes the following source data and figure supplement(s) for figure 8:

**Source data 1.** Raw numerical data for *Figure 8B, C, and E–H*.

**Source data 2.** PDF file containing original colonic tissue image for *Figure 8D*.

**Source data 3.** Original files for colonic tissue image displayed in *Figure 8D*.

**Figure supplement 1.** The effect of prophylactic *B. velezensis* HBXN2020 on the bacterial load in feces or ileum of mice and the histological score of the colons.

**Figure supplement 1—source data 1.** Raw numerical data for *Figure 8—figure supplement 1A–C*.

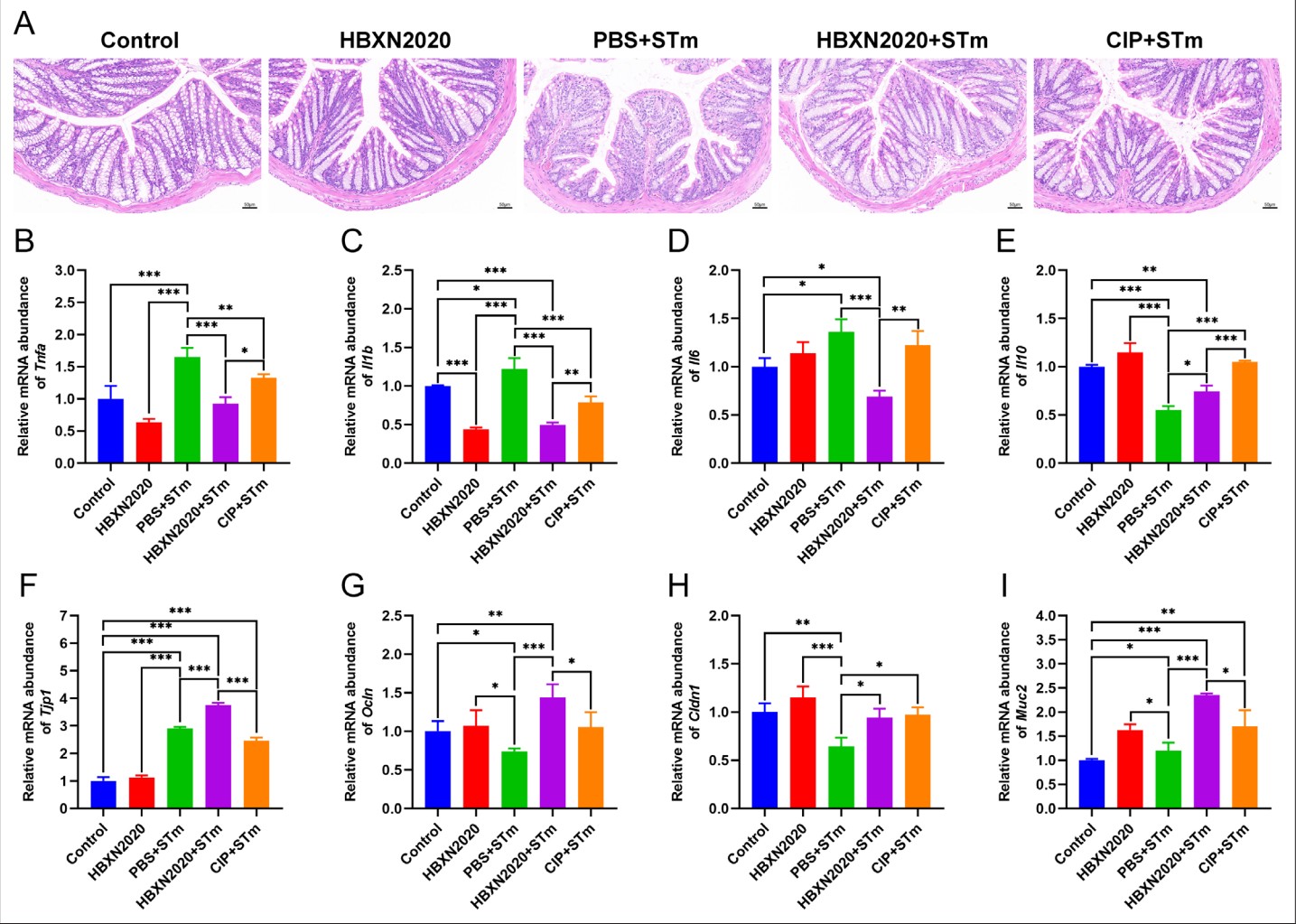

**Figure 9.** Prophylactic *B. velezensis* HBXN2020 spores attenuated colonic damage and inflammatory reaction. (**A**) Hematoxylin and eosin (H&E) stained colon tissue sections. Scale bar: 50 μm. The mRNA levels of (**B**) *Tnfa*, (**C**) *Il1b*, (**D**) *Il6*, and (**E**) *Il10* were detected by RT-qPCR. Data were shown as mean values ± SEM (n=5). The mRNA levels of (**F**) *Tjp1*, (**G**) *Ocln*, (**H**) *Cldn1*, and (**I**) *Muc2* in colon tissue were detected by RT-qPCR. Data were shown as mean values ± SEM (n=5). Statistical significance was evaluated using one-way analysis of variance (ANOVA) with Tukey's multiple comparisons test (\*, p<0.05, \*\*, p<0.01, and \*\*\*, p<0.001).

The online version of this article includes the following source data for figure 9:

**Source data 1.** Raw numerical data for *Figure 9B–I*.

## Discussion

*S. Typhimurium* is an important intestinal pathogen that can cause invasive intestinal diseases such as bacterial colitis (*Herp et al., 2019*; *Schultz et al., 2017*). Antibiotics are frequently used to treat colitis caused by bacteria; nevertheless, in recent years, their abuse has greatly increased bacterial resistance, leading to serious environmental pollution. A recent study showed that antibiotic resistance genes in probiotics could be transmitted to the intestinal microbiota, which may threaten human health (*Crits-Christoph et al., 2022*). It is crucial to consider the source of probiotics. Strong environmental adaptability and disease resistance make black pigs an outstanding household breed in China (*Yang et al., 2022*). In this study, *B. velezensis* HBXN2020 was isolated from the free-range feces of black piglets in a mountain village in Xianning City (Hubei, China) and exhibited excellent antibacterial activity. Otherwise, in vitro tolerance assays showed that *B. velezensis* HBXN2020 spores have good tolerance to high temperature, strong acids, bile salts, as well as simulated gastric and intestinal fluid, which was similar to previous research results (*Du et al., 2022*). Furthermore, studies for antibiotic susceptibility revealed that *B. velezensis* HBXN2020 is not resistant to antibiotics.

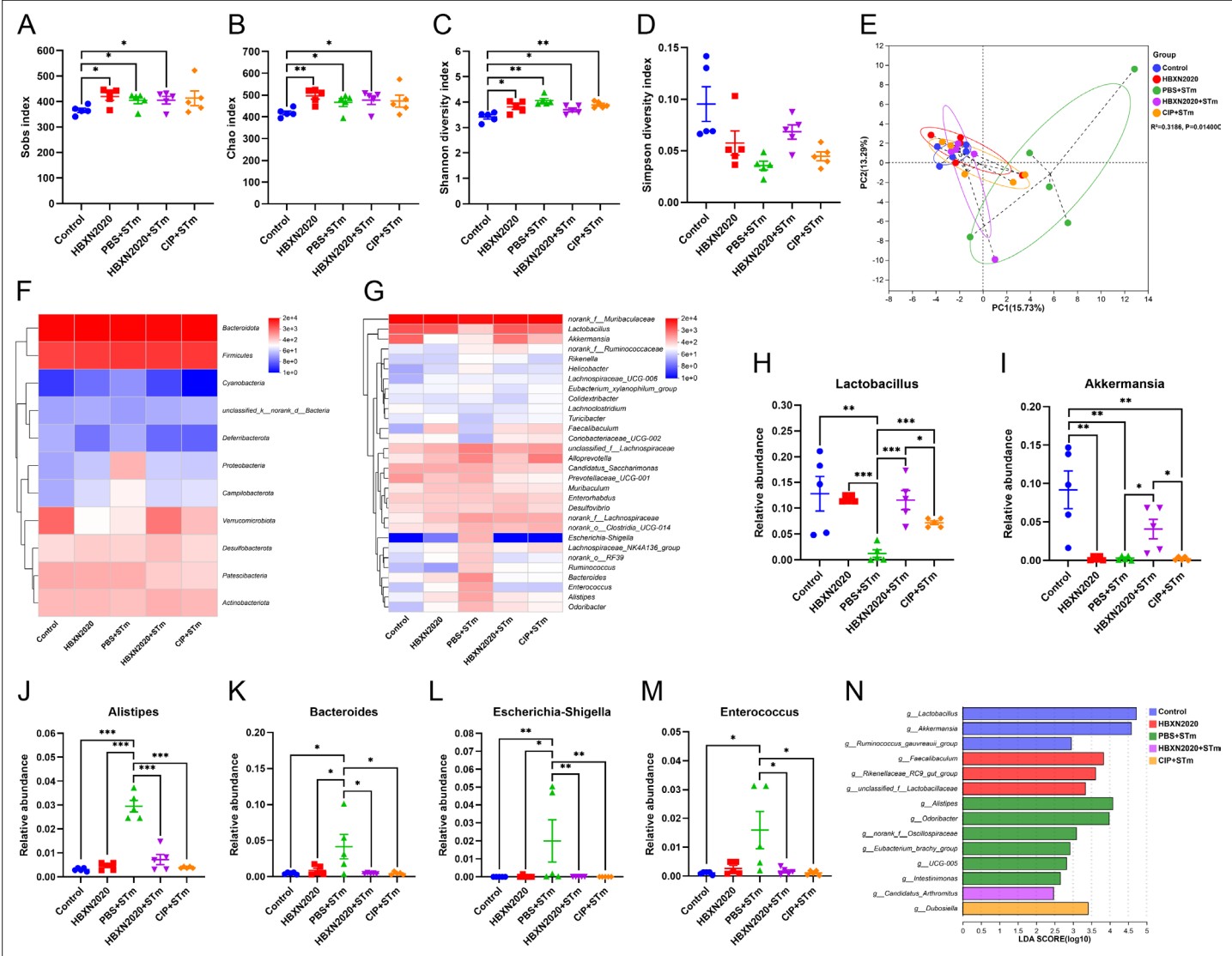

**Figure 10.** Prophylactic *B. velezensis* HBXN2020 spores regulated the composition of gut microbiota. (**A–D**) Alpha diversity of the intestinal microbiota. Data were shown as mean values ± SEM (n=5). (**E**) The principal components analysis (PCA) plot showed the β diversity among different microbial community groups based on Bray-Curtis distance at the operational taxonomic unit (OTU) level. Heatmap of the community composition of colonic microbiota at the phylum (top 15 phyla) (**F**) and genus (top 30 genera) (**G**) level. Relative abundance of selected taxa (**H**) *Lactobacillus*, (**I**) *Akkermansia*, (**J**) *Alistipes*, (**K**) *Bacteroides*, (**L**) *Escherichia-Shigella*, and (**M**) *Enterococcus.* Data were shown as mean values ± SEM (n=5). (**N**) Analysis of differences in the microbial taxa by LEfSe (linear discriminant analysis [LDA] effect size) (LDA score>2) in different groups. Significance was evaluated by one-way analysis of variance (ANOVA) with Tukey's multiple comparisons test (*, p<0.05, **, p<0.01, and ***, p<0.001).

The online version of this article includes the following source data and figure supplement(s) for figure 10:

**Source data 1.** Raw numerical data for *Figure 10A–D and H–M*.

**Figure supplement 1.** Analysis of the community compositions of colonic microbiota.

Probiotics are distinct from food or medications in that they can cause toxins or infections in the body when ingested, and they are living when swallowed (*Sanders et al., 2010*; *Snydman, 2008*). Therefore, assessing the biosafety of probiotics is crucial. In this study, the safety of *B. velezensis* HBXN2020 was evaluated by measuring the body weight, intestinal barrier proteins, and inflammatory cytokines of the experimented mice, as well as conducting routine blood and biochemical tests. The results showed that the expression levels of *Tjp1* and *Ocln* in the colon of mice increased after treatment with different doses of *B. velezensis* HBXN2020 spores. It has been reported that *Tjp1*, *Ocln*, and *Cldn1* are the three most important tight junction proteins in intercellular connections, which play

a crucial role in maintaining the intestinal epithelial barrier (*Bazzoni et al., 2000*; *Zihni et al., 2016*). Thus, it is possible to improve intestinal barrier function by upregulating *Tjp1* and *Ocln* expression levels.

Important aminotransferases in animals, AST and ALT, are regarded as critical metrics for assessing liver function damage (*Gou et al., 2022*; *Ozer et al., 2008*). Under normal circumstances, the levels of ALT and AST are in dynamic balance without notable changes. However, when the liver is damaged or becomes dysfunctional, the levels of aminotransferases (ALT and AST) increase significantly (*Tang et al., 2012*). Previous studies have shown that CCL4-induced liver injury strongly increases plasma ALT/AST levels (*Singhal et al., 2018*). Zhang et al. reported that the addition of *Bacillus subtilis* to chicken diets significantly decreased the serum levels of ALT and AST (*Zhang et al., 2017*). In this present study, the levels of ALT and AST in the probiotic-treated group were slightly lower than those in the control group, indicating that *B. velezensis* HBXN2020 had no negative effect on liver health in mice. The *B. velezensis* HBXN2020 treatment group's blood indices were also found to be comparable to those of the control group, falling within the normal reference range, according to regular blood testing.

Mice were fed with *B. velezensis* HBXN2020 spores orally after being challenged with *S.* Typhimurium ATCC14028 in order to assess the spores' potential to alleviate infection caused by *S.* Typhimurium. Our experimental results demonstrated that oral treatment with *B. velezensis* HBXN2020 spores could alleviate infection by *S.* Typhimurium, as evidenced by decreased weight loss, DAI, and histological damage, which is similar to prior studies (*Cao et al., 2019*; *Wu et al., 2022*). Previous research showed that macrophages are the first line of host defense against bacterial infection. They release proinflammatory cytokines, which are critical in initiating adaptive immune responses (*Cheng et al., 2014*; *Dinarello, 2000*). Nevertheless, proinflammatory cytokines have immunological properties that can be beneficial for the host to resist the invasion of bacteria and other microbes in the surrounding environment (*Liu et al., 2021*). For instance, pretreatment with recombinant murine *Tnfa* was shown to protect mice against lethal bacterial (*E. coli*) infection (*Cross et al., 1989*). PJ-34 exerted protective effects on intestinal epithelial cells against invasive *Salmonella* infection by upregulating *Il6* expression through the ERK and NF-κB signaling pathways (*Huang, 2009*).

On the other hand, some research has demonstrated that over-activation of immune cells can result in tissue damage, organ failure, systemic or persistent inflammation, and autoimmune diseases (*Karki et al., 2021*; *Kotas and Medzhitov, 2015*). *Il10* is a recognized anti-inflammatory mediator that plays a crucial role in maintaining intestinal microbe-immune homeostasis, regulating the release of inflammatory mediators, and inhibiting proinflammatory responses of innate and adaptive immunity (*Maloy and Powrie, 2011*; *Ouyang et al., 2011*; *Saraiva et al., 2020*). For instance, previous studies have shown that *Clostridium butyrate* can induce the production of *Il10* in the intestine, thereby alleviating experimental colitis in mice (*Hayashi et al., 2013*). Similar results were observed in our study, where oral administration of *B. velezensis* HBXN2020 spores reduced the expression levels of *Tnfa*, *Il1b*, and *Il6* while increasing the levels of *Il10* in the colon of mice. Additionally, as demonstrated by the decreased histological damage and increased expression levels of intestinal barrier proteins, oral treatment of *B. velezensis* HBXN2020 spores reduced the functional damage of the intestinal barrier induced by S. Typhimurium infection.

Besides host itself, environmental factors such as diet and gut microbiota have been associated with the development of STm-infected. Gut microbiota constitutes a critical bridge between environmental factors and host health, where the beneficial bacteria such as *Lactobacillus* in the microbiota may exert repair and anti-inflammatory function (*Li et al., 2022*; *Liu et al., 2022*). Moreover, supplementation of probiotics has been shown to modulate intestinal microbiota and reduce the risk of STm-infected (*Buddhasiri et al., 2021*; *Zhang et al., 2022*). This is consistent with our research results between the *B. velezensis* HBXN2020 treatment group and STm-infected mice. Here, we found that oral *B. velezensis* HBXN2020 spores raised the colonic microbiota's alpha diversity and modulated the composition of the intestinal microbiota, namely, enriching the relative abundance of *Lactobacillus* (known for enhancing the epithelial barrier by increasing mucus secretion and upregulating the expression of tight junction proteins such as *Cldn1*, *Ocln*, and *Tjp1*) (*Kaur et al., 2021*) and *Akkermansia* (known for modulating intestinal immune response by producing short-chain fatty acids and reducing the secretion of proinflammatory cytokines) (*Cani et al., 2022*), reducing the relative abundance of *Enterococcus*, *Salmonella*, and *Bacteroides* (harmful to intestinal homeostasis) (*Wang*

*et al., 2023*). Notably, we also found that the relative abundance of *Lactobacillus* and *Akkermansia* in the ciprofloxacin treatment group was significantly lower than that in the *B. velezensis* HBXN2020 spores treatment group, indicating that excessive use of antibiotics may interfere with the recovery of beneficial bacteria in the gut. Collectively, infection with *S.* Typhimurium disrupts gut microbiota and the gut barrier, leading to intestinal inflammation, while oral *B. velezensis* HBXN2020 spores protects the colon and reduce infection by increasing the abundance of beneficial bacteria and barrier integrity and decreasing inflammation.

We further assessed pretreatment in order to learn more about the potential of *B. velezensis* HBXN2020 spores for disease prevention, given the increased efficacy of treating infection caused by *S.* Typhimurium. Similar to the therapeutic effect of oral *B. velezensis* HBXN2020, prophylactic *B. velezensis* HBXN2020 spores effectively alleviated the symptoms of STm-infected mice, namely, reduced the mRNA levels of proinflammatory cytokines and increased the anti-inflammatory cytokine level in the colon of STm-infected mice. Prophylactic *B. velezensis* HBXN2020 spores also increased the mRNA levels of gut barrier proteins in the colon of STm-infected mice. Furthermore, the microbiota composition was also dramatically modulated by prophylactic *B. velezensis* HBXN2020 spores in both healthy mice and STm-infected mice. Also, the community richness revealed by the alpha diversity index were not significantly impacted in STm-infected mice by prophylactic *B. velezensis* HBXN2020 spores. Interestingly, we found that prophylactic *B. velezensis* HBXN2020 spores significantly increased the community diversity in healthy mice, suggesting the potentially beneficial effects of *B. velezensis* HBXN2020 in gut microenvironment for animals in the future. Moreover, prophylactic *B. velezensis* HBXN2020 spores significantly boosted SCFA-producing bacteria including *Lactobacillus* and *Akkermansia*, which was consistent with the effects of therapeutic oral *B. velezensis* HBXN2020 spores. Meanwhile, harmful bacteria such as *Alistipes*, *Bacteroides*, *Escherichia-Shigella*, and *Enterococcus* were significantly enriched in STm-infected mice, reminding us that increased harmful bacteria might play a major role in promoting the development of STm-infected. In summary, our results demonstrated that *B. velezensis* HBXN2020-mediated microbial communities, especially *Lactobacillus* and *Akkermansia*, might play a crucial role in alleviating the development of STm-infected.

## Conclusion

In conclusion, *S.* Typhimurium ATCC14028 infection can result in gut barrier disruption and intestinal inflammation and intestinal microbiota dysbiosis. Supplementing *B. velezensis* HBXN2020 spores can improve intestinal microbiota stability and gut barrier integrity and reduce inflammation to help prevent or treat infection by *S.* Typhimurium. Thus, our results indicate that supplementing *B. velezensis* HBXN2020 may represent a novel option for preventing *Salmonella* infections.

**Key resources table**

| Reagent type (species) or resource | Designation | Source or reference | Identifiers | Additional information |
|---|---|---|---|---|
| Strain, strain background (*Bacillus velezensis*) | HBXN2020 | This paper | NCBI GenBank accession No: CP119399.1 | This strain is used in the entire text |
| Strain, strain background (*Salmonella* Typhimurium) | STm | ATCC | 14028 | |
| Strain, strain background (*Escherichia coli*) | 25922 | ATCC | 25922 | |
| Strain, strain background (*E. coli*) | 35150 | ATCC | 35150 | |
| Strain, strain background (*E. coli*) | EC024 | This paper | | This strain is used in *Figure 2B* |
| Strain, strain background (*S.* Typhimurium) | SL1344 | *Gao et al., 2022* | | |
| Strain, strain background (*S.* Enteritidis) | SE006 | *Gao et al., 2022* | | |

*Continued on next page*

*Continued*

| Reagent type (species) or resource | Designation | Source or reference | Identifiers | Additional information |
|---|---|---|---|---|
| Strain, strain background (*Staphylococcus aureus*) | 29213 | ATCC | 29213 | |
| Strain, strain background (*S. aureus*) | 43300 | ATCC | 43300 | |
| Strain, strain background (*S. aureus*) | S21 | This paper | | This strain is used in *Figure 2B* |
| Strain, strain background (*Clostridium perfringens*) | 2030 | CVCC | 2030 | |
| Strain, strain background (*C. perfringens*) | CP023 | This paper | | This strain is used in *Figure 2B* |
| Strain, strain background (*C. perfringens*) | CP002 | This paper | | This strain is used in *Figure 2B* |
| Strain, strain background (*Streptococcus suis*) | SC19 | *Duan et al., 2023* | | |
| Strain, strain background (*S. suis*) | SS006 | This paper | | This strain is used in *Figure 2B* |
| Strain, strain background (*Pasteurella multocida*) | PM002 | This paper | | This strain is used in *Figure 2B* |
| Strain, strain background (*P. multocida*) | PM008 | This paper | | This strain is used in *Figure 2B* |
| Strain, strain background (*Actinobacillus pleuropneumoniae*) | APP015 | This paper | | This strain is used in *Figure 2B* |
| Strain, strain background (*A. pleuropneumoniae*) | APP017 | This paper | | This strain is used in *Figure 2B* |
| Sequence-based reagent | *Tnfa*-F | This paper | qPCR primers | CCACGCTCTTCTGTCTACTG |
| Sequence-based reagent | *Tnfa*-R | This paper | qPCR primers | ACTTGGTGGTTTGCTACGA |
| Sequence-based reagent | *Il1b*-F | This paper | qPCR primers | ACCTGTGTCTTTCCCGTGG |
| Sequence-based reagent | *Il1b*-R | This paper | qPCR primers | TCATCTCGGAGCCTGTAGTG |
| Sequence-based reagent | *Il6*-F | This paper | qPCR primers | GAGCCCACCAAGAACGATA |
| Sequence-based reagent | *Il6*-R | This paper | qPCR primers | TTGTCACCAGCATCAGTCC |
| Sequence-based reagent | *Il10*-F | This paper | qPCR primers | TGGACAACATACTGCTAACCG |
| Sequence-based reagent | *Il10*-R | This paper | qPCR primers | GGGCATCACTTCTACCAGGT |
| Sequence-based reagent | *Tjp1*-F | This paper | qPCR primers | CTGGTGAAGTCTCGGAAAAATG |
| Sequence-based reagent | *Tjp1*-R | This paper | qPCR primers | CATCTCTTGCTGCCAAACTATC |
| Sequence-based reagent | *Ocln*-F | This paper | qPCR primers | CAGGATGCCAATTACCATCAAG |
| Sequence-based reagent | *Ocln*-R | This paper | qPCR primers | GGGTTCACTCCCATTATGTACA |
| Sequence-based reagent | *Cldn1*-F | This paper | qPCR primers | AGATACAGTGCAAAGTCTTCGA |
| Sequence-based reagent | *Cldn1*-R | This paper | qPCR primers | CAGGATGCCAATTACCATCAAG |
| Sequence-based reagent | *Muc2*-F | This paper | qPCR primers | CGAGCACATCACCTACCACATCATC |
| Sequence-based reagent | *Muc2*-R | This paper | qPCR primers | TCCAGAATCCAGCCAGCCAGTC |
| Sequence-based reagent | β-actin-F | This paper | qPCR primers | GACCTCTATGCCAACACAGT |
| Sequence-based reagent | β-actin-R | This paper | qPCR primers | CACCAATCCACACAGAGTAC |

*Continued on next page*

*Continued*

| Reagent type (species) or resource | Designation | Source or reference | Identifiers | Additional information |
|---|---|---|---|---|
| Commercial assay or kit | HiScript III RT SuperMix | Vazyme Biotechnology Co., Ltd | #R323-01 | |
| Commercial assay or kit | qPCR SYBR Green Master Mix | Yeasen Biotechnology Co., Ltd | #11203ES08 | |
| Commercial assay or kit | E.Z-N.A Stool DNA Kit | Omega | #D4015-01 | |
| Chemical compound, drug | Kanamycin | Solarbio | #K8020 | |
| Chemical compound, drug | Ciprofloxacin | Solarbio | #C9710 | |
| Chemical compound, drug | Ampicillin | Hangzhou Binhe Microorganism Reagent Co., Ltd | #C002 | 10 μg/tablet |
| Chemical compound, drug | Meropenem | Hangzhou Binhe Microorganism Reagent Co., Ltd | #C102 | 10 μg/tablet |
| Chemical compound, drug | Piperacillin | Hangzhou Binhe Microorganism Reagent Co., Ltd | #C005 | 10 μg/tablet |
| Chemical compound, drug | Gentamycin | Hangzhou Binhe Microorganism Reagent Co., Ltd | #C017 | 10 μg/tablet |
| Chemical compound, drug | Tetracycline | Hangzhou Binhe Microorganism Reagent Co., Ltd | #C021 | 30 μg/tablet |
| Chemical compound, drug | Doxycycline | Hangzhou Binhe Microorganism Reagent Co., Ltd | #C032 | 30 μg/tablet |
| Chemical compound, drug | Minocycline | Hangzhou Binhe Microorganism Reagent Co., Ltd | #C046 | 30 μg/tablet |
| Chemical compound, drug | Erythromycin | Hangzhou Binhe Microorganism Reagent Co., Ltd | #C023 | 15 μg/tablet |
| Chemical compound, drug | Enrofloxacin | Shunyou Shanghai Biotechnology Co., Ltd | #CT0639B | 5 μg/tablet |
| Chemical compound, drug | Ofloxacin | Hangzhou Binhe Microorganism Reagent Co., Ltd | #C044 | 5 μg/tablet |
| Chemical compound, drug | Sulfamethoxazole | Shunyou Shanghai Biotechnology Co., Ltd | #CT0051B | 25 μg/tablet |
| Chemical compound, drug | Trimethoprim-sulfamethoxazole | Hangzhou Binhe Microorganism Reagent Co., Ltd | #C027 | 23.75 μg/tablet |
| Chemical compound, drug | Polymyxin B | Shunyou Shanghai Biotechnology Co., Ltd | #CT0044B | 300 U/tablet |
| Chemical compound, drug | Teicoplanin | Shunyou Shanghai Biotechnology Co., Ltd | #CT0647B | 30 μg/tablet |
| Chemical compound, drug | Trimethoprim | Shunyou Shanghai Biotechnology Co., Ltd | #CT0076B | 5 μg/tablet |
| Chemical compound, drug | Florfenicol | Shunyou Shanghai Biotechnology Co., Ltd | #CT1754B | 30 μg/tablet |

*Continued on next page*

*Continued*

| Reagent type (species) or resource | Designation | Source or reference | Identifiers | Additional information |
|---|---|---|---|---|
| Chemical compound, drug | Spectinomycin | Shunyou Shanghai Biotechnology Co., Ltd | #CT0046B | 10 µg/tablet |
| Chemical compound, drug | Nitrofurantoin | Shunyou Shanghai Biotechnology Co., Ltd | #CT0036B | 300 µg/tablet |
| Chemical compound, drug | Rifampicin | Hangzhou Binhe Microorganism Reagent Co., Ltd | #C013 | 5 µg/tablet |

# Materials and methods

## Sample collection and strain isolation

Fresh fecal samples were collected from the farms of various locations in southern China including Hubei, Anhui, Hunan, Jiangxi, Guizhou, and Guangxi. All samples were placed in sterile plastic bags, sealed and placed on ice, and immediately transported to the laboratory. *Bacillus* was isolated following a previously described method with slight modification (*Unban et al., 2020*). The obtained strains were kept in Luria-Bertani (LB) media containing 25% (vol/vol) glycerol and stored at −80°C, and prepared in LB medium or LB agar medium before use.

## Antibacterial activity of *Bacillus*

The antibacterial activity of *Bacillus* isolates was evaluated by using the spot-on-plate method on LB agar plates supplemented with indicator bacteria (*Salmonella*, *E. coli*, *S. aureus*). Briefly, an overnight culture of indicator bacteria was dipped by sterile cotton swabs and spread on LB agar plates. *Bacillus* solution was spotted onto the double-layer agar plates and incubated at 37°C for 12 hr to measure the inhibition zone. A transparent zone of at least 1 mm around the spot was considered positive.

The antibacterial spectrum of *Bacillus* isolates was compared through the agar diffusion test, based on the results of the spot-on-plate test. In brief, an overnight culture of indicator strains was mixed with 10 mL of TSB soft agar (TSB broth containing 0.7% [wt/vol] agar) and poured into a sterile plate covered with LB agar and Oxford cups, 8 mm diameter wells were prepared in the TSB agar after removing the cups. The wells were filled with 100 µL of *Bacillus* solution. The plates were incubated at 37°C for 14 hr, and the inhibition zone was measured.

The indicator strains used in this study are listed in *Supplementary file 3*, and all the primers used in *Supplementary file 3*. *C. perfringens* were cultured in a fluid thioglycollate medium (FTG) (Hopebio, Qingdao, China) at 45°C. *S. suis* and *P. multocida* were cultured in TSB medium supplemented with 5% (vol/vol) sheep serum (Solarbio, Beijing, China) at 37°C, and *A. pleuropneumoniae* were cultured in TSB supplemented with 5% (vol/vol) sheep serum and 5 mM nicotinamide adenine dinucleotide at 37°C, while the other bacterial strains were cultured in LB medium (Solarbio, Beijing, China) at 37°C. In addition, the Difco sporulation medium was used for inducing the sporulation of *Bacillus* via the nutrient depletion method (*Tang et al., 2017*).

## Growth curves of HBXN2020

The growth curve of HBXN2020 was recorded in flat-bottomed 100-well microtiter plates via detecting optical density at 600 nm ($OD_{600}$) at 1 hr intervals using the automatic growth curve analyzer (Bioscreen, Helsinki, Finland).

## In vitro resistance assay of HBXN2020

HBXN2020 spores (100 µL) or vegetative cells (100 µL) were separately resuspended in 900 µL of LB medium supplemented with different pH values (2, 3, 4, 5, or 6), bile salts (0.85% NaCl, 0.3%), simulated gastric fluid (SGF, HCl, pH 1.2) containing 10 g/L of pepsin in 0.85% NaCl solution, or simulated intestinal fluid (SIF, NaOH, pH 6.8) containing 10 g/L of trypsin in 0.05 M $KH_2PO_4$ solution, and incubated at 37°C. A normal LB medium (pH 7.0) was used as the control. At predetermined time points, 100 µL was taken from each sample, serially diluted 10-fold with sterile PBS (pH 7.2), and then spread onto LB agar plates. The plates were incubated overnight in a constant temperature incubator at

37°C, and the bacterial colonies were counted. The survival rate was calculated using the following formula: Survival rate = (number of bacteria in the treatment group/number of bacteria in the control group)×100%.

One milliliter of HBXN2020 spores or vegetative cells were separately placed in water baths at different temperatures (37°C, 45°C, 55°C, 65°C, 75°C, 85°C, or 95°C) for 20 min, with a 37°C water bath used as the control. The survival rate was calculated as described above.

## Antibiotic susceptibility assays

Antimicrobial susceptibility testing was performed using the Kirby-Bauer disk diffusion method in accordance with the Clinical Laboratory Standards Institute (CLSI) guidelines (*CLSI, 2018*.). The antimicrobial agents tested were: Ampicillin, Meropenem, Piperacillin, Gentamycin, Tetracycline, Doxycycline, Minocycline, Erythromycin, Enrofloxacin, Ofloxacin, Sulfamethoxazole, Trimethoprim-sulfamethoxazole, Polymyxin B, Teicoplanin, Trimethoprim, Florfenicol, Spectinomycin, Nitrofuran-toin, and Rifampicin. The diameter of the inhibition zone was measured using a vernier caliper.

## Antimicrobial assays

The in vitro antagonistic activity of HBXN2020-CFS was tested using the agar well-diffusion method against 18 indicator strains (pathogens), which included 7 standard strains (*E. coli* ATCC 25922, *E. coli* ATCC 35150, *S.* Typhimurium ATCC 14028 (STm), *S.* Typhimurium SL1344, *S. aureus* ATCC 29213, *S. aureus* ATCC 43300, and *C. perfringens* CVCC 2030) and 11 clinical isolates (*E. coli* EC024, *S.* Enteritidis SE006, *S. aureus* S21, *C. perfringens* CP023, *C. perfringens* CP002, *S. suis* SC19, *S. suis* SS006, *P. multocida* PM002, *P. multocida* PM008, *A. pleuropneumoniae* APP015, and *A. pleuropneumoniae* APP017). All plates were cultured at 37°C for 16 hr before observing the inhibition zone, and the diameter of the inhibition zone was measured using a vernier caliper. The presence of a clear zone indicated antagonistic activity.

## In vitro bacterial competition

To investigate whether HBXN2020 directly inhibited the growth of STm, we performed spot-on lawn and agar well-diffusion assays, as well as co-culture assays in liquid culture medium. In the spot-on lawn antimicrobial assays, we prepared double layers of agar by first pouring LB agar into the plate as the bottom layer. The top layer consisted of 10 mL TSB broth containing 0.7% agar with STm overnight culture. Then, 10 μL of HBXN2020 overnight culture and cell-free supernatant (CFS) were respectively spotted onto TSB agar and incubated at 37°C for 12 hr to measure the inhibition zone. A transparent zone of at least 1 mm around the spot was considered positive.

The antagonistic effect of HBXN2020-CFS against STm was determined using the agar well-diffusion assays. To collect HBXN2020-CFS, the culture was centrifuged at 9000×*g* for 15 min at 4°C, and the supernatant was filtered through a 0.22 μm membrane filter (Millipore, USA). The STm lawn medium was prepared by mixing 10 mL of TSB broth containing 0.7% agar with STm overnight culture and then poured into a sterile plate covered with LB agar and Oxford cups, 8 mm diameter wells were prepared in the TSB agar after removing the cups. The wells were filled with 100 μL of HBXN2020-CFS, LB medium, or ampicillin (100 μg/mL). The plates were incubated at 37°C for 14 hr, and the inhibition zone was measured.

The co-culture assay was conducted by incubating HBXN2020 and *S.* Typhimurium ATCC14028 (carry pET28a (+), kanamycin resistance) (*Cao et al., 2019*) separately overnight at 37°C and diluting them to $10^4$ CFU/mL. Then, the two strains were mixed at different ratios (1:1, 1:10, 1:50, or 1:100) and co-cultured at 37°C with shaking (180 rpm). At predetermined time points, serial 10-fold dilutions were prepared for all samples and spread onto selective (kanamycin 50 μg/mL) LB agar plates and cultured at 37°C for 12 hr before bacterial counting. Viable colony counts ranged from 30 to 300 per plate.

## Genome sequencing, annotation, and analysis

The complete genome of strain was sequenced using the Illumina HiSeq PE and PacBio RSII (SMRT) platforms (Shanghai Majorbio Bio-Pharm Technology Co., Ltd.). Briefly, the short reads from the Illumina HiSeq PE were assembled into contigs using SPAdenovo (http://soap.genomics.org.cn/). The long reads from the PacBio RSII and the Illumina contigs were then aligned using the miniasm and

Racon tools in Unicycler (version 0.4.8, https://github.com/rrwick/Unicycler, copy archived at *Wick, 2024*) to generate long-read sequences. During the assembly process, sequence correction was performed using Pilon (version 1.22, https://github.com/broadinstitute/pilon/wiki/Standard-Output; https://github.com/broadinstitute/pilon copy archived at *Walker, 2021*). Lastly, a complete genome with seamless chromosomes was obtained.

The CDSs of the strain genome were predicted using Glimmer (version 3.02, http://ccb.jhu.edu/software/glimmer/index.shtml). The tRNA and rRNA genes were predicted using TRNAscan-SE (version 2.0, http://trna.ucsc.edu/software/) and Barrnap (version 0.8, https://github.com/tseemann/barrnap, copy archived at *Seemann, 2019*), respectively. The functional annotation of all CDSs was performed using various databases, including Swiss-Prot Database (https://web.expasy.org/docs/swiss-prot_guideline.html), Pfam Database (http://pfam.xfam.org/), EggNOG Database (http://eggnog.embl.de/), GO Database (http://www.geneontology.org/), and KEGG Database (http://www.genome.jp/kegg/). The circular map of the strain genome was generated using CGView (version 2, http://wishart.biology.ualberta.ca/cgview/) (*Stothard and Wishart, 2005*).

## Comparative genomic analysis

To elucidate the phylogenetic relationships of the strain from a whole-genome perspective, a phylogenetic tree based on the whole genome was constructed using the Type (Strain) Genome Server (TYGS) online (https://ggdc.dsmz.de/) (*Meier-Kolthoff et al., 2022*). The ANI values of the strain genome to the reference strain genome were then calculated using the JSpeciesWS online service (*Richter et al., 2016*), and a heatmap was generated using TBtools (version 1.113) (*Chen et al., 2020*).

## In vivo safety evaluation of *B. velezensis* HBXN2020

Female specific-pathogen-free (SPF) C57BL/6 mice aged 6–8 weeks were purchased from the Animal Experimental Center of Huazhong Agricultural University. All mice experiments were conducted in the standard SPF facility of Huazhong Agricultural University, with 12 hr of light and 12 hr of darkness at a temperature of 25°C and ad libitum access to food and water. The use of animals in this experiment was approved by the Animal Care and Ethics Committee of Huazhong Agricultural University (Ethics Approval Number: HZAUMO-2023-0089).

Safety assay of *B. velezensis* HBXN2020 referred to the method described by *Zhou et al., 2022*, with slight modifications. After a 7-day acclimation period (free access to water and food), mice were randomly divided into four treatment groups (n=5): low-dose *B. velezensis* HBXN2020 spores group (L-HBXN2020 group, $10^7$ CFU/mouse), medium-dose *B. velezensis* HBXN2020 spores group (M-HBXN2020 group, $10^8$ CFU/mouse), high-dose *B. velezensis* HBXN2020 spores group (H-HBXN2020 group, $10^9$ CFU/mouse), and a control group. During the experimental period (15 days), mice were weighed and orally gavaged with their respective treatments once every 2 days. On day 15, all mice were euthanized, and blood, heart, liver, spleen, lung, kidney, ileum, cecum, and colon were collected. Blood samples were used for routine blood and biochemistry tests. Major organ tissues (heart, liver, spleen, lung, and kidney) and a 5 mm distal segment of the colon were used for histopathology. The remaining colonic tissues were rapidly frozen in liquid nitrogen and stored at −80°C for cytokine and tight junction protein expression analysis.

## *S.* Typhimurium-infected mouse model

After 7 days of acclimation (free access to water and food), mice were randomly divided into five treatment groups (n=6): control group, HBXN2020 group, STm+PBS group, STm+HBXN2020 group, and STm+CIP group. The *S.* Typhimurium-infected model was performed as previously described (*Stecher et al., 2005*), with slight modifications. On the first day of the experiment, all mice in the STm+PBS group, STm+HBXN2020 group, and STm+CIP group were orally inoculated with 200 µL ($5×10^7$ CFU/mouse) of STm. On days 1, 3, and 5 following STm infection, each mouse in the HBXN2020 group, STm+HBXN2020 group, and STm+CIP group received 200 µL ($1×10^8$ CFU/mouse) of *B. velezensis* HBXN2020 spores or ciprofloxacin (1 mg/mL) via gavage administration, respectively. In contrast, the control group and STm+PBS group received 200 µL of sterile PBS by oral gavage. Fecal samples were collected daily following STm infection and resuspended in sterile PBS. The number of STm in mice feces from both the STm+PBS, STm+HBXN2020, and STm+CIP groups was then determined by spreading a serial 10-fold dilution on selective LB agar plates containing 50 µg/mL kanamycin.

Throughout the entire experiment, the body weight, stool consistency, and fecal occult blood of all mice were monitored daily. As shown in *Supplementary file 3*, DAI was calculated by the sum of the scores from three parameters (*Praveschotinunt et al., 2019*). On day 7 after STm infection, all mice were euthanized, their ileum, cecum, and colon were collected. The length of colon was measured, and a 5 mm distal segment of the colon was fixed in 4% paraformaldehyde for further histopathology. The remaining colon was then rapidly frozen in liquid nitrogen and stored at −80°C for cytokine and tight junction protein expression analysis. Then, the number of STm in the ileum, cecum, and colon was determined by spreading serial 10-fold dilutions on selective LB agar plates.

To investigate the prophylactic efficacy of *B. velezensis* HBXN2020 in ameliorating STm-infected, another independent experiment was conducted using 6-week-old female C57BL/6 mice (SPF). After a 7-day acclimation period, mice were randomly assigned to five groups (n=6): control group, HBXN2020 group, PBS +STm group, HBXN2020+STm group, and CIP+STm group. On days 1, 3, 5, and 7, each mouse in the HBXN2020 group, HBXN2020+STm group, and CIP+STm group received 200 μL (1×10$^8$ CFU/mouse) of *B. velezensis* HBXN2020 spores or ciprofloxacin (1 mg/mL) via gavage administration. Meanwhile, mice in the control group and PBS+STm group received 200 μL of sterile PBS by oral gavage. On day 7, all mice in the PBS+STm group, HBXN2020+STm group, and CIP+STm group were orally inoculated with 200 μL (5×10$^7$ CFU/mouse) of STm. Fecal samples were collected daily following STm infection from both groups and resuspended in sterile PBS. The number of STm in the feces of mice was then determined by spreading serial 10-fold dilutions on selective LB agar plates (50 μg/mL kanamycin). Throughout the entire experiment, the body weight and DAI scores of all mice were monitored daily. On day 12, all mice were euthanized, and the ileum, cecum, and colon were collected. The colon length was measured, and a 5 mm distal segment of the colon was fixed in 4% formalin for sectioning and staining. The remaining colon was stored at −80°C for future analysis. Lastly, the number of STm in the ileum, cecum, and colon was determined using a selective LB agar plate.

## Determination of cytokines and tight junction protein expression in colon tissue

Total RNA was extracted from colon tissues using TRIpure reagent (Aidlab, China), and cDNA was obtained using HiScript III RT SuperMix for qPCR (+gDNA wiper) (Vazyme, China). RT-qPCR for each gene was performed in triplicate using qPCR SYBR Green Master Mix (Yeasen Biotechnology, Shanghai, China). The relative expression level of cytokine and tight junction protein genes was calculated using the 2$^{-\Delta\Delta Ct}$ method with β-actin and GAPDH as reference genes. The primer sequences used in the RT-qPCR test are listed in *Supplementary file 3*.

### Histopathology analysis

Colon tissue samples (0.5 cm) were fixed in 4% paraformaldehyde for 24 hr, and the fixed tissues were then embedded in paraffin and sectioned. The sections were stained with hematoxylin and eosin (H&E) and observed and imaged using an optical microscope (Olympus Optical, Tokyo, Japan). The histopathological score included the degree of inflammatory infiltration, changes in crypt structures, and the presence or absence of ulceration and edema. The scoring criteria were determined as previously described (*Wu et al., 2022*).

### 16S rRNA gene sequencing and analysis

According to the manufacturer's instructions, colon microbial community genomes were extracted using E.Z.N.A Stool DNA Kit (Omega; D4015-01), and quality was detected by 1% agarose gel electrophoresis. The 16S rRNA V3-V4 variable region was amplified by PCR using universal primers 338F (5 '- ACTCCTACGGGGGGCAG-3') and 806R (5 '- GACTACHVGGGTWTCTAAT-3'). The PCR products were examined by electrophoresis on 2% agarose gels and then purified with the AxyPrep DNA gel extraction kit (Axygen Biosciences, USA). A sequencing library was constructed using the NEXT-FLEX Rapid DNA-Seq Kit, and sequencing was performed using the Illumina MiSeq platform. Raw reads were quality evaluated and filtered by fastp (version 0.20.0) and merged using FLASH (version 1.2.7). The optimized sequences were clustered into operational taxonomic units (OTUs) based on 97% sequence similarity using UPARSE (version 7.1). The representative sequences of each OTU was classified by RDP classifier (version 2.2; confidence threshold value, 0.7). Alpha diversity was assessed

using the ACE, Chao, Shannon, and Simpson indices. The β diversity analysis was performed using Bray-Curtis distances and visualized through PCA. LefSe was used to identify differential microbiota between groups.

## Statistical analysis

Statistical analysis was performed using GraphPad Prism 8.3.0 (GraphPad Software, San Diego, CA, USA) and Excel (Microsoft, Redmond, WA, USA). Data are presented as mean ± standard error of the mean (SEM). Differences between two groups were evaluated using two-tailed unpaired Student's t-test, and all other comparisons were conducted using one-way analysis of variance (ANOVA) with Tukey's multiple comparisons test. For all analyses, significance differences are denoted as: *, $p<0.05$, **, $p<0.01$, and ***, $p<0.001$.

## Acknowledgements

This work was supported by grants from the National Program on Key Research Project of China (2021YFD1800300, 2022YFD1800800), 'Yingzi Tech & Huazhong Agricultural University Intelligent Research Institute of Food Health' (No. IRIFH202209; IRIFH202301), and the Fundamental Research Funds for the Central Universities (2662016PY004).

RT-qPCR data were acquired at the National Key Laboratory of Agricultural Microbiology Core Facility.

## Additional information

### Funding

| Funder | Grant reference number | Author |
| --- | --- | --- |
| National Program on Key Research Project of China | 2021YFD1800300 | Ping Qian |
| Yingzi Tech Huazhong Agricultural University Intelligent Research Institute of Food Health | IRIFH202209 | Ping Qian |
| Fundamental Research Funds for the Central Universities | 2662016PY004 | Ping Qian |
| National Program on Key Research Project of China | 2022YFD1800800 | Ping Qian |
| Yingzi Tech Huazhong Agricultural University Intelligent Research Institute of Food Health | IRIFH202301 | Ping Qian |

The funders had no role in study design, data collection and interpretation, or the decision to submit the work for publication.

### Author contributions

Linkang Wang, Conceptualization, Data curation, Formal analysis, Methodology, Writing – original draft; Haiyan Wang, Xinxin Li, Mengyuan Zhu, Data curation, Methodology; Dongyang Gao, Dayue Hu, Zhixuan Xiong, Methodology, Writing - review and editing; Xiangmin Li, Supervision, Writing - review and editing; Ping Qian, Conceptualization, Supervision, Funding acquisition, Project administration, Writing - review and editing

### Author ORCIDs

Ping Qian (iD) https://orcid.org/0000-0003-3120-2556

### Ethics

The use of all animals in this experiment was approved by the Animal Care and Ethics Committee of Huazhong Agricultural University (Ethics Approval Number: HZAUMO-2023-0089).

Reviewer #1 (Public review): https://doi.org/10.7554/eLife.93423.4.sa1
Author response https://doi.org/10.7554/eLife.93423.4.sa2

## Additional files

### Supplementary files
- Supplementary file 1. Determination of antibacterial activity of different *Bacillus.*
- Supplementary file 2. HBXN2020 genome features and clusters of secondary metabolic synthesis genes. (a) HBXN2020 genome features. (b) Clusters of secondary metabolic synthesis genes in HBXN2020.
- Supplementary file 3. The bacterial strains, RT-qPCR primers, and disease activity index scoring schemes in this study. (a) The bacterial strains used in this study. (b) The primer sequences used for the RT-qPCR. (c) Disease activity index (DAI) parameters and their associated scoring schemes.
- MDAR checklist

### Data availability
Whole genome sequencing data of Bacillus velezensis have been deposited in NCBI GenBank under accession code CP119399.1.16s rRNA sequencing data of mice have been deposited in NCBI under the accession code PRJNA1150571.

The following datasets were generated:

| Author(s) | Year | Dataset title | Dataset URL | Database and Identifier |
|---|---|---|---|---|
| Wang LK, Qian P | 2023 | Bacillus velezensis strain HBXN2020 chromosome, complete genome | https://www.ncbi.nlm.nih.gov/nuccore/CP119399.1?report=genbank&to=3929792 | NCBI GenBank, CP119399.1 |
| Wang LK | 2024 | Effect of probiotic Bacillus on the gut microbiota of mice | http://www.ncbi.nlm.nih.gov/bioproject/PRJNA1150571 | NCBI BioProject, PRJNA1150571 |

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
