## [Editor Report · eLife Assessment]

In this **useful** study, Wang and colleagues investigate the potential probiotic effects of Bacillus velezensis in a murine model. They provide **convincing** evidence that B. velezensis limits the growth of *Salmonella* typhimurium in lab culture and in mice, together with beneficial effects on the microbiota. The overall presentation of the manuscript has improved and the work will be of interest to infectious disease researchers.

---

## [Referee Report · Reviewer #1 (Public review)]

Summary:

Wang and colleagues presented an investigation of pig-origin bacteria Bacillus velezensis HBXN2020, for its released genome sequence, in vivo safety issue, probiotic effects in vitro, and protection against *Salmonella* infection in a murine model. Various techniques and assays are performed; the main results are all descriptive, without new insight advancing the field or a mechanistic understanding of the observed protection.

Strengths:

An extensive study on the probiotic properties of the Bacillus velezensis strain HBXN2020

Weaknesses:

The main results are descriptive without mechanistic insight. Additionally, most of the results and analysis parts are separated without a link or a story-telling way to deliver a concise message.

Now the manuscript has made appropriate and considerable improvements.

---

## [Author Response]

The following is the authors’ response to the previous reviews.

**Reviewer #1 (Public Review):**
Summary:Wang and colleagues presented an investigation of pig-origin bacteria Bacillus velezensis HBXN2020, for its released genome sequence, in vivo safety issue, probiotic effects in vitro, and protection against *Salmonella* infection in a murine model. Various techniques and assays are performed; the main results are all descriptive, without new insight advancing the field or a mechanistic understanding of the observed protection.

Thank you very much for your reading and comments our manuscript.

Strengths:An extensive study on probiotic property of the Bacillus velezensis strain HBXN2020

Thank you very much for your reading and comments our manuscript.

Weaknesses:The main results are descriptive without mechanistic insight. Additionally, most of the results and analysis parts are separated without a link or a story-telling way to deliver a concise message.

Thank you for your comments and suggestions on our manuscript. In later work, we will focus on exploring the antibacterial substances and bactericidal mechanisms of *B. velezensis*. The manuscript results and analysis sections have been extensively revised. We appreciate your review and feedback.

**Reviewer #2 (Public Review):**
Summary:In this study, Wang and colleagues study the potential probiotic effects of Bacillus velezensis. Bacillus species have potential benefit to serve as probiotics due to their ability to form endospores and synthesize secondary metabolites. B. velezensis has been shown to have probiotic effects in plants and animals but data for human use are scarce, particularly with respect to *Salmonella*-induced colitis. In this work, the authors identify a strain of B. velezensis and test it for its ability to control colitis in mice.

Thanks for the constructive comments and the positive reception of the manuscript.

Key findings:(1) The authors sequence an isolate for B. velezensis - HBXN2020 and describe its genome (roughly 4 mb, 46% GC-content etc).

Thanks for the constructive comments and the positive reception of the manuscript.

(2) The authors next describe the growth of this strain in broth culture and survival under acid and temperature stress. The susceptibility of HBXN2020 was tested against various antibiotics and against various pathogenic bacteria. In the case of the latter, the authors set out to determine if HBXN2020 could directly inhibit the growth of pathogenic bacteria. Convincing data, indicating that this is indeed the case, are presented.

Thanks for the constructive comments and the positive reception of the manuscript.

(3) To determine the safety profile of BHXN2020 (for possible use as a probiotic), the authors infected the strain in mice and monitored weight, together with cytokine profiles. Infected mice displayed no significant weight loss and expression of inflammatory cytokines remained unchanged. Blood cell profiles of infected mice were consistent with that of uninfected mice. No significant differences in tissues, including the colon were observed.

Thanks for the constructive comments and the positive reception of the manuscript.

(4) Next, the authors tested the ability to HBXN2020 to inhibit growth of *Salmonella* typhimurium (STm) and demonstrate that HBXN2020 inhibits STm in a dose dependent manner. Following this, the authors infect mice with STm to induce colitis and measure the ability of HBXN2020 to control colitis. The first outcome measure was a reduction in STm in faeces. Consistent with this, HBXN2020 reduced STm loads in the ileum, cecum, and colon. Colon length was also affected by HBXN2020 treatment. In addition, treatment with HBXN2020 reduced the appearance colon pathological features associated with colitis, together with a reduction in inflammatory cytokines.

Thanks for the constructive comments and the positive reception of the manuscript.

(5) After noting the beneficial (and anti-inflammatory effects) of HBXN2020, the authors set out to investigate effects on microbiota during treatment. Using a variety of algorithms, the authors demonstrate that upon HXBN2020 treatment, microbiota composition is restored to levels akin to that seen in healthy mice.

Thanks for the constructive comments and the positive reception of the manuscript.

(6) Finally, the authors assessed the effect of using HBXN2020 as prophylactic treatment for colitis by first treating mice with the spores and then infecting with STm. Their data indicate that treatment with HBXN2020 reduced colitis. A similar beneficial impact was seen with the gut microbiota.

Thanks for the constructive comments and the positive reception of the manuscript.

Strengths:(1) Good use of in vitro and animal models to demonstrate a beneficial probiotic effect.

Thank you very much for your reading and comments our manuscript.

(2) Most observations are supported using multiple approaches.

Thanks for the comments and the positive reception of the manuscript.

(3) Mouse experiments are very convincing.

Thanks for the comments and the positive reception of the manuscript.

Weaknesses:(1) Whilst a beneficial effect is observed, there no investigation of the mechanism that underpins this.

Thank you for pointing this out. We apologize for any inconvenience caused by the lack of mechanism research of the manuscript. In later work, we will focus on exploring the antibacterial substances and bactericidal mechanisms of *B. velezensis*. Thank you for your suggestions, and we hope our response has addressed your concerns.

(2) Mouse experiments would have benefited from the use of standard anti-inflammatory therapies to control colitis. That way the authors could compare their approach of using bacillus spores that current gold standard for treatment.

We gratefully appreciate for your valuable comments. The comments improve the quality and depth of manuscript. Based on your suggestion, we have supplemented this in the revised manuscript. We appreciate your review and feedback, and have marked the updated contents in the revised manuscript.

The updated contents were presented in line 198-378 in results section of the revised manuscript.

**Reviewer #3 (Public Review):**
Summary:The manuscript by Wang et al. investigates the effects of B. velezensis HBXN2020 in alleviating S. Typhimurium-induced mouse colitis. The results showed that B. velezensis HBXN2020 could alleviate bacterial colitis by enhancing intestinal homeostasis (decreasing harmful bacteria and enhancing the abundance of Lactobacillus and Akkermansia) and gut barrier integrity and reducing inflammation.

Thanks for the comments and the positive reception of the manuscript.

Strengths:B. velezensis HBXN2020 is a novel species of Bacillus that can produce a great variety of secondary metabolites and exhibit high antibacterial activity against several pathogens. B. velezensis HBXN2020 is able to form endospores and has strong anti-stress capabilities. B. velezensis HBXN2020 has a synergistic effect with other beneficial microorganisms, which can improve intestinal homeostasis.

Thanks for the comments and the positive reception of the manuscript.

Weaknesses:Few studies about the clinical application of Bacillus velezensis. Thus, more studies are still needed to explore the effectiveness of Bacillus velezensis before clinical application.

Thanks for your suggestion. This study serves as an exploratory investigation before the application of *Bacillus velezensis*. The main purpose of this study is to explore the potential of *Bacillus velezensis* in application. We appreciate your review and feedback and hope that our response adequately addresses your concerns.

**Recommendations for the authors:**

**Reviewer #1 (Recommendations For The Authors):**
Most of my previous comments are well addressed, here are a few examples.https://pubs.rsc.org/en/content/articlelanding/2020/fo/d0fo01017k please replace "colitis" to a normal infection model. The current statement is incorrect.While in my last comment, I requested a Colitis Mouse Model, which will well resemble the diarrhea disease caused by *Salmonella* in mammals. The available statement is not convincing, please check https://www.ncbi.nlm.nih.gov/pmc/articles/PMC2225501/,

Thank you for your valuable suggestion. The comments improve the quality of manuscript. We have corrected this in the revised manuscript as suggested. We have marked the updated contents in the revised manuscript.

The updated contents were presented in line 2, 29, 38, 46, 48, 199, 204, 246, 248, 282, 307, 310, 316, 431, 433, 464, 466, 473, 494, 497, 499, 504, 513, 518, 525, 706, 710 and 735 in the revised manuscript.

Certain parts remain to be overestimated, to my knowledge, the language and logical flow should be addressed thoroughly.Here are suggestions to improve the logical flow of the manuscript.(1) Probiotic sampling and isolation(2) in vitro assessment(3) genomic sequencing and in silico safety assessment (Crit Rev Food Sci Nutr. 2023;63(32):11244-11262), which should be included as a right ref.(4) in vivo assay for safety evaluation, but not biosafety (it has a different meaning!!)(5) infection model and protection assay.

We gratefully appreciate for your valuable comments. The comments improve the quality and depth of manuscript. According to your suggestion, we do our best to correct those problems in the revised manuscript. We would like to express our apologies once again and hope that the revised manuscript meets your expectations. We have marked the updated contents in the revised manuscript.

Also, please pay attention to the logical link or transition sentences between each part to connect the dots in each part.

We gratefully appreciate for your valuable comments. The comments improve the quality of manuscript. According to your suggestion, we have corrected this in the revised manuscript. We have marked the updated contents in the revised manuscript.

Finally, there are also lots of typos and errors, please improve through the text.For example, Line 521. "Stain", and more...

Thanks for pointing this out. Based on your suggestion, we have corrected in the revised manuscript. We have marked the updated contents in the revised manuscript.

The updated contents were presented in line 753, 1055, 1087 in the revised manuscript.

**Reviewer #2 (Recommendations For The Authors):**
The revised manuscript by Wang and colleagues attempts to address concerns raised during the first round of review.All minor comments have been addressed and in general, the major concerns have been partially addressed in the revised manuscript.The outstanding concerns relate to the mechanistic basis of the observations. The authors made no attempt to address this in a meaningful manner. Secondly, the issue of comparing the responses to what would be standard therapy (such as anti-inflammatories) was also handled in a somewhat dismissive manner, referring to other ongoing/future work. The clinical utility of the findings are hard to ascertain if there is no comparison to the current gold standard therapeutic approach.I have no further suggestions for the authors, save for those previously made.

Thank you for pointing this out. We apologize for any inconvenience caused by the lack of mechanism research of the manuscript. In later work, we will focus on exploring the antibacterial substances and bactericidal mechanisms of *B. velezensis*. Thank you for your suggestions, and we hope our response has addressed your concerns.

Secondly, About the comparative trial of oral *bacillus* spore treatment with the current gold standard for treatment, we have supplemented this in the revised manuscript. We appreciate your review and feedback, and have marked the updated contents in the revised manuscript.

The updated contents were presented in line 198-378 in results section of the revised manuscript.

**Reviewer #3 (Recommendations For The Authors):**
This is a revision, they have addressed all my concerns, and now it is acceptable.

Thank you very much for your comments and recognition of the manuscript.